# Pretrain Value, Not Reward: Decoupled Value Policy Optimization

**Chenghua Huang**♦,*, **Lu Wang**♣1, **Fangkai Yang**♣, **Pu Zhao**♣,
**Qingwei Lin**♣, **Dongmei Zhang**♣, **Saravan Rajmohan**♣
♦School of Computer Science, Fudan University
♣Microsoft
huangch22@m.fudan.edu.cn
{wlu,fangkaiyang,puzhao,dongmeiz,saravar}@microsoft.com

## Abstract

In this paper, we explore how directly pretraining a value model simplifies and stabilizes reinforcement learning from human feedback (RLHF). In reinforcement learning, value estimation is the key to policy optimization, distinct from reward supervision. The value function predicts the *return-to-go* of a partial answer, that is, how promising the partial answer is if it were continued to completion. In RLHF, however, the standard pipeline first pretrains a reward model and then learns a value function online, even though no new reward signals are available once preference data is collected. This makes critic learning redundant, as the process of training a reward model and then deriving a value model is informationally equivalent to directly pretraining a value model. Importantly, this requires no additional supervision, and our value model is trained on exactly the same data used for reward modeling. Building on this insight, we introduce *Decoupled Value Policy Optimization* (DVPO), a framework that pretrains a *Global Value Model* (GVM) offline and freezes it as a universal critic for policy learning. The GVM provides stable, fine-grained credit assignment without critic drift or trajectory sampling. Experiments across MT-Bench, Alpaca-Eval, and Arena-Hard demonstrate that DVPO matches or surpasses state-of-the-art RLHF methods. These results highlight RLHF can be reframed as policy-only optimization guided by a single pretrained value model. The implementation code for our method is available in https://github.com/microsoft/DKI_LLM/tree/main/dvpo

## 1 Introduction

The alignment of large language models (LLMs) with human preferences has become one of the core challenges of modern natural language processing. While scaling up LLMs has led to remarkable gains in performance and generality Achiam et al. (2023); Bubeck et al. (2023), scale alone does not ensure that models behave in ways users actually desire, especially in complex reasoning, summarization, or dialogue tasks. Reinforcement Learning from Human Feedback (RLHF) Christiano et al. (2017); Bai et al. (2022b); Song et al. (2024) has emerged as a critical step in shaping model behavior, enabling systems that are safer, more accurate, and more useful (Bai et al., 2022a; Ganguli et al., 2022).

A central component of RLHF is reinforcement learning (RL), where policies are optimized with respect to *value estimates* of return-to-go. Existing RLHF frameworks, however, have taken a circuitous path toward value estimation. First, because LLMs lack an interactive environment that provides ground-truth rewards, the community trains a *reward model* (RM) from preference data (Ziegler et al., 2019; Ouyang et al., 2022; Wang et al., 2024). Second, this reward model is used to supervise either an online-trained critic (as in PPO (Schulman et al., 2017) and its variants) or to indirectly estimate values through trajectory sampling (as in DPO (Rafailov et al., 2024), ReMax (Li et al., 2023), and GRPO (Shao et al., 2024)). Both approaches are costly and unstable: critics drift

---

*Work is done during an internship at Microsoft
1Corresponding author.

during joint training, while sampling-based methods discard token-level credit assignment and suffer from high variance. (Yao et al., 2023; Huang et al., 2024)

This work explores a simpler and more direct alternative, motivated by two ideas. First, if no new ground-truth rewards are available during policy training, then learning a value model online from a fixed reward model adds no fundamentally new information. Pretraining a reward model followed by critic training is therefore informationally equivalent to directly pretraining a value model. Second, in open-ended tasks, rewards are largely policy-invariant: whether policy A or policy B produces a solution, the return is determined by correctness or preference, not by policy-specific stochasticity. This opens the door to amortizing value estimation into a *Global Value Model* (GVM), pretrained once on diverse trajectories and reused as a frozen critic across policies.

We call this approach **Decoupled Value Policy Optimization (DVPO)**. DVPO eliminates online critic training entirely: the GVM is pretrained offline to predict token-level return-to-go, frozen, and then used to guide policy optimization via a standard RL objective. Crucially, the GVM is trained on exactly the same preference data that would ordinarily be used to pretrain a reward model, requiring no additional supervision or annotations. This design avoids critic drift, reduces compute and memory overhead, and provides fine-grained interpretability through token-level attributions.

We present empirical evaluations across standard RLHF benchmarks, including MT-Bench Zheng et al. (2023), Alpaca-Eval Dubois et al. (2024), and Arena-Hard Li et al. (2024). Our results show that DVPO achieves performance on par with or surpassing state-of-the-art RLHF methods while reducing GPU memory usage by 30-40% and training time by 30-45%. This indicates that rather than repeatedly retraining unstable actor–critic pairs or discarding value functions altogether, a single offline-pretrained value model can serve as a stable, scalable, and generalizable critic for RLHF.

## 2  RELATED WORK

**Reinforcement Learning in Language Model Optimization.** Reinforcement learning has emerged as a prevalent method for fine-tuning large language models (LLMs), with Proximal Policy Optimization (Schulman et al., 2017) and its variations (Ramamurthy et al., 2022; Wu et al., 2023) being the most prevalent methods. These techniques largely adhere to the actor-critic paradigm (Sutton, 2018), This approach alternates between training a value estimator for the current policy and leveraging it to enhance policy performance. This bilevel process may result in a suboptimal policy, as demonstrated by empirical studies (Gao et al., 2023). Moreover, the alternating optimization of the policy and critic models, along with the use of rewards provided by the reward model as environmental feedback, necessitates loading four models (including the reference model) simultaneously during training. This significantly increases training complexity and computational resource consumption (Yao et al., 2023; Hu et al., 2024).

**Efficiency RLHF.** Many recent studies have sought to mitigate the computational complexity and resource consumption of the reinforcement learning (RL) step in RLHF. Methods such as DPO (Rafailov et al., 2024) and its variants (Meng et al., 2024; Ethayarajh et al., 2024; Hong et al., 2024) bypass reward modeling and the actor-critic learning framework by directly learning from preferences. However, existing research indicates that due to their offline nature, these approaches exhibit a performance gap compared to online RL (Xu et al., 2024). Some recent works have proposed a reward-only approach to reduce the training cost of the RL phase (Li et al., 2023; Gunter et al., 2024; Shao et al., 2024; Ahmadian et al., 2024; Yu et al., 2025). However, this method lacks value estimation and assigns the same reward score to each token, leading to high variance and instability during training (Hu, 2025). Unlike these approaches, our method pre-trains a global value model (GVM) and leverages it to guide RL training, providing token-level supervision signals. This not only reduces training resource consumption but also stabilizes the training process, achieving performance comparable to the original PPO.

Some recent works have attempted to learn a value function and use it to guide the decoding phase of LLMs, thereby bypassing the RL optimization stage (Han et al., 2024; Kong et al., 2024; Mao et al., 2024). However, this approach significantly increases inference complexity and raises inference costs. Some studies (Yuan et al., 2025) also leverage pre-trained value models to guide policy optimization; however, they still employ an actor–critic architecture, incurring substantial computational overhead.

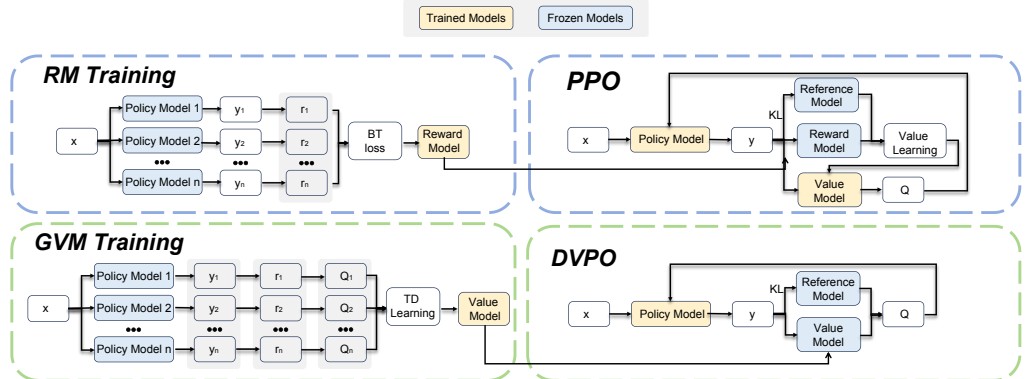

Figure 1: **Overview of DVPO.** PPO requires a pre-trained reward model to provide environmental feedback. It combines this reward signal with a learned value function to estimate the action–value $Q(s, a)$. In contrast, DVPO trains a Global Value Model (GVM) on the same offline data to produce fine-grained value estimates. This design removes the need for additional ground-truth rewards during policy training and decouples policy learning from value learning.

In contrast, we leverage the learned value model to guide RL training, where the pre-trained value model helps the policy model converge more stably (Noukhovitch et al., 2024).

## 3 METHOD

We propose DVPO for RLHF in which a GVM is trained once and then fixed to guide policy updates. This approach removes the need for joint policy–value training and mitigates the associated computational overhead and instability. As shown in Figure 1, our method comprises two primary stages: (1) **Train GVM:** Use offline trajectories (states, actions, returns, and policy-specific data) to learn a policy-conditioned action-value function $Q_\phi$. Notably, returns can be derived from either preference data—commonly used in standard RLHF settings—or a pretrained reward model. For a detailed comparison of these sources, see Appendix G. (2) **Decoupled Value Policy Optimization:** Freeze $Q_\phi$ and optimize a policy using a standard RL objective (e.g., PPO), taking advantage estimates from the fixed GVM.

### 3.1 PROBLEM SETTING

We model the sequence generation task in NLP as a Markov Decision Process (MDP). The response consists of $T$ tokens, denoted by $y = y^{<T+1} \coloneqq [y^1, y^2, \ldots, y^T]$, where $y^t \in \mathcal{Y}$ and $\mathcal{Y}$ is the vocabulary. We assume $y^{<1} = [\,]$, indicating an empty prefix. Given a prompt $x$ and the first $t-1$ tokens $y^{<t}$, the language model (LM) predicts the probability distribution for the next token as: $\pi_\theta(\cdot \mid [x, y^{<t}])$.

In this MDP formulation, the state is defined as $s_t = [x, y^{<t}]$, representing the prompt (i.e., $x$) and the generated response up to step $t$. The action is the next generated token $a_t = y^t$. The ground-truth reward at the sentence level, denoted by $r(x, y)$, is provided by human feedback. To transform the sentence-level reward into token-level rewards $r(s_t, a_t)$, we apply a simplified form of temporal difference (TD) learning. Specifically, we assign a reward of 0 to each intermediate step and use the final step's reward as the sentence-level reward value.

By this definition, the cumulative return from step $t$ onwards is expressed as: $G_t = \sum_{k=t}^{T} \gamma^{k-t} r(s_k, a_k)$, which simplifies to $G_t = \gamma^{T-t} r(x, y)$ when all intermediate rewards are zero. In RLHF, we rely on offline data $\mathcal{D}$, which contains state–action trajectories, returns, and policy behaviors. The dataset is defined as: $\mathcal{D} = \left\{ (\tau_i, s_t^i, a_t^i, G_t^i) \right\}_{i=1}^{N}$, where $\tau_i$ represents the $i$-th policy behavior, typically expressed as a sequence of question–answer pairs, $(s_t^i, a_t^i)$ is a state–action pair sampled under that policy. Note that $\tau_i$ and $(s_t^i, a_t^i)$ may refer to different or identical question–answer pairs under the same policy. No additional ground-truth rewards are collected during training.

## 3.2 TRAINING A GLOBAL VALUE MODEL (GVM)

Our key insight is to learn a *policy-conditioned* action-value function $Q_\phi(\tau, s, a)$ from offline data, where $\tau$ represents a sampled trajectory capturing *how the policy behaves* in unrelated contexts. This conditioning approximates how well a particular policy, embodied by $\tau$, would perform when taking action $a$ in state $s$.

**Policy Conditioning via Trajectories.** Traditional actor–critic methods require online adaptation of the value function to the actor's evolving behavior. In contrast, we aim for a single, *global* $Q_\phi$ that generalizes across different policies, thus avoiding iterative re-learning. Instead of conditioning on explicit policy parameters, we leverage *trajectories* $\tau$ randomly sampled from the policy in question. Each $\tau$ is a sequence of question–answer pairs (in LLM tasks) or other interactions that reveal distinct policy traits (e.g., stylistic tendencies, correctness, domain expertise).

Formally, we parametrize: $Q_\phi(\tau, s, a) \approx \mathbb{E}\Big[\sum_{t=0}^{\infty} \gamma^t r(s_t, a_t) \ \Big| \ s_0 = s, \ a_0 = a, \ \tau\Big]$, where $\tau$ implicitly determines which policy $\pi(\cdot \mid s)$ we are approximating. In practice, we train the global value model $Q_\phi(\tau, s, a)$ using Temporal Difference (TD) learning. The target return $G_i$ is estimated based on future rewards and value predictions. Specifically, $G_i$ is computed as:

$$G_t = r(s_t, a_t) + \gamma Q_\phi(\tau, s_{t+1}, a_{t+1}),$$

where $r(s_t, a_t)$ is the reward for taking action $a_t$ in state $s_t$, and $Q_\phi(\tau, s_{t+1}, a_{t+1})$ is the predicted value of the next state-action pair.

The training objective for the global value model is to minimize the TD loss:

$$\mathcal{L}_{\text{GVM}}(\phi) \ = \ \mathbb{E}_{(\tau_t, s_t, a_t, r_t, s_{t+1}, a_{t+1}) \in \mathcal{D}}\big[\big(r_t + \gamma Q_\phi(\tau, s_{t+1}, a_{t+1}) - Q_\phi(\tau, s_t, a_t)\big)^2\big], \quad (1)$$

This TD-based loss drives $Q_\phi$ to iteratively adjust its estimates of the expected return, ensuring that the value function accurately reflects both immediate and future rewards. This approach is particularly suitable for offline RLHF scenarios, as it can handle sequences with deterministic state transitions.

## 3.3 DECOUPLED-VALUE POLICY OPTIMIZATION

In traditional actor–critic methods, both the policy (actor) and value (critic) are trained simultaneously, which introduces instability due to their interdependence. This issue is exacerbated in offline RLHF settings, where no new environment rewards can be collected to correct misaligned updates. To address these challenges, we propose a decoupled-value policy optimization approach: the global value model $Q_\phi$ is trained and fixed before policy optimization, decoupling the learning dynamics of the actor and critic.

**Policy Optimization Objective.** Once the global value model $Q_\phi$ converges, we *fix* its parameters and use it to guide policy updates. Let $\pi_\theta(a \mid s)$ be the policy to be optimized. We adopt a clipped PPO objective, which stabilizes policy updates by limiting the impact of large policy changes:

$$\mathcal{L}_{\text{PPO}}(\theta) \ = \ \mathbb{E}\Big[\min\big(r_t(\theta)\hat{A}_t, \text{clip}\big(r_t(\theta), 1 - \epsilon, 1 + \epsilon\big)\hat{A}_t\big)\Big], \quad (2)$$

where $r_t(\theta) = \frac{\pi_\theta(a_t|s_t)}{\pi_{\theta_{\text{old}}}(a_t|s_t)}$ is the importance sampling ratio, and $\epsilon$ is a clipping parameter to prevent overly large updates.

The advantage function measures the relative quality of an action compared to the expected value of the state. We define the advantage using the fixed value model:

$$\hat{A}_t = \widetilde{Q}_\phi(\tau, s_t, a_t),$$

where $\widetilde{Q}_\phi$ is the value estimate computed during the GVM training stage (see §3.2). In offline RLHF, this static advantage definition provides a reliable signal for policy improvement without requiring dynamic value updates.

Our decoupled-value approach offers two key advantages: (1) By fixing $Q_\phi$, we eliminate the "moving target" problem inherent in actor–critic methods, leading to more stable and predictable policy updates.

(2) Since no additional environment rewards can be collected, the static $Q_\phi$ provides all necessary supervisory information, enabling efficient reuse of the offline dataset for policy optimization.

In addition, DVPO does not require stronger assumptions than standard PPO-style RLHF. Because the GVM is pretrained once and fixed, policy optimization proceeds with stable guidance and avoids critic drift. We provide empirical evidence of this stability in Appendix F, where DVPO exhibits smooth and reliable training curves compared to PPO.

### 3.4 EQUIVALENCE OF REWARD–CRITIC TRAINING AND VALUE PRETRAINING

In RLHF with fixed feedback, we first train a reward model $R_\phi$ on a preference dataset $\mathcal{D}$. Existing pipelines then *estimate values from this trained reward*—either by learning an online critic from $R_\phi$ (PPO-style) or by sampling many outputs and normalizing $R_\phi$ scores . Crucially, both routes consume no supervision beyond $\mathcal{D}$: the value signal is entirely derived from the already-trained $R_\phi$. Therefore, the two-stage procedure *(reward training) + (value estimation from the trained reward)* is **informationally equivalent** to *directly pretraining a value model $Q_\psi$ on the same $\mathcal{D}$*. This is the equivalence we claim: not that rewards and values are identical, but that *deriving a value signal from a fixed, pretrained reward adds no new information compared to pretraining the value model directly*. DVPO leverages this by pretraining a Global Value Model once and discarding the redundant estimation stage.

#### 3.4.1 PRELIMINARIES

Let $r(s, a)$ denote the unknown ground-truth reward function, and let $\gamma \in [0, 1]$ be the discount factor. We assume access to an offline dataset $\mathcal{D}$, from which we pretrain the following models. (1) **Reward Model** $R_\phi(s, a)$, which approximates $r(s, a)$ with error bounded by $\epsilon_R$. (2) **Global Value Model** $Q_\psi(s, a)$, which approximates the action-value function $Q^\pi(s, a) = \mathbb{E}_\pi \left[ \sum_{t=0}^\infty \gamma^t r(s_t, a_t) \mid s_0 = s, a_0 = a \right]$, with approximation error bounded by $\epsilon_Q$.

We merge any explicit conditioning (e.g., trajectory $\tau$) into the state definition to simplify notation. Since no new rewards are observed post-deployment, both $R_\phi$ and $Q_\psi$ serve as static supervision signals throughout training.

Let $\pi_\theta(a \mid s)$ be the trainable policy. We define the policy gradient objectives under each supervision signal as:

$$\nabla_\theta \mathcal{J}_R(\pi_\theta) = \mathbb{E}_{s \sim d^{\pi_\theta}, a \sim \pi_\theta} \left[ \nabla_\theta \log \pi_\theta(a \mid s) \cdot \hat{A}_R(s, a) \right],$$

$$\nabla_\theta \mathcal{J}_Q(\pi_\theta) = \mathbb{E}_{s \sim d^{\pi_\theta}, a \sim \pi_\theta} \left[ \nabla_\theta \log \pi_\theta(a \mid s) \cdot \hat{A}_Q(s, a) \right],$$

where $d^{\pi_\theta}$ is the stationary distribution under policy $\pi_\theta$, and the advantage functions are defined as:

$$\hat{A}_R(s, a) = \widetilde{Q}_\phi^R(s, a) - b(s), \quad \hat{A}_Q(s, a) = Q_\psi(s, a) - b(s),$$

with $b(s)$ as a baseline (e.g., state value function $V(s)$) to reduce variance. The estimate $\widetilde{Q}_\phi^R(s, a)$ is computed by solving the Bellman equation using $R_\phi$, following standard value iteration procedures. This structure parallels typical actor-critic methods, as discussed in (Sutton, 2018). In contrast, $Q_\psi(s, a)$ is a *fixed global value model* pretrained offline. This is a key distinction: unlike standard actor-critic methods (e.g., PPO), where the value function is updated alongside the policy at each training step.

#### 3.4.2 EQUIVALENCE LEMMA.

We formalize the observation that, under fixed preference data, (reward pretraining + value estimation) provides no additional information beyond direct value pretraining.

**Lemma 3.1** (Equivalence under fixed feedback). *Let $\mathcal{D}$ be a fixed preference dataset and let $R_\phi$ be a reward model trained on $\mathcal{D}$. Assume: 1. For all $(s, a)$ in $\mathcal{D}$, $|R_\phi(s, a) - r(s, a)| \leq \epsilon_R$. 2. For the same distribution, both the reward-induced value $\widetilde{Q}_\phi^R$ and the pretrained GVM $Q_\psi$ satisfy*

$$\left| \widetilde{Q}_\phi^R(s, a) - Q^\pi(s, a) \right| \leq \epsilon_Q, \qquad \left| Q_\psi(s, a) - Q^\pi(s, a) \right| \leq \epsilon_Q.$$

*3. (Fixed feedback) No new rewards are observed during training. Then the induced policy gradients satisfy*

$$\left\| \nabla_\theta \mathcal{J}_{\widetilde{Q}_\phi^R}(\pi_\theta) - \nabla_\theta \mathcal{J}_{Q_\psi}(\pi_\theta) \right\| \leq \kappa(\epsilon_R, \epsilon_Q),$$

*with $\kappa(\epsilon_R, \epsilon_Q) \to 0$ as $\epsilon_R, \epsilon_Q \to 0$. Thus, reward–critic training and direct value pretraining are* informationally equivalent *on $\mathcal{D}$.*

The complete proof, including Bellman-based derivations and bounding arguments, is provided in Appendix B.

**Convergence Corollary.** Building on Lemma 3.1, we now consider DVPO policy updates. If policy updates are regularized (e.g., KL-clipped as in PPO) and the GVM approximation error is bounded, then DVPO inherits the same monotonic improvement guarantee as PPO. In other words, replacing the online critic with a fixed pretrained GVM does not weaken PPO's local convergence properties, while eliminating critic drift in practice.

# 4 EXPERIMENT

## 4.1 TRAINING DATASETS

*Stage1*: We collected offline trajectories data to train GVM. Specifically, we selected the *UltraFeedback* dataset(Cui et al., 2023), in which, for a given prompt, multiple models generate responses that are annotated with scalar scores. Training instances for GVM take the form $\{\text{prompt}, \text{response}_i, \text{reward}_i\}$. We also explored training directly from pairwise preferences by assigning a score of $1$ to the chosen response and $-1$ to the rejected response. The resulting performance was comparable to training with scalar rewards, highlighting GVM's compatibility with different supervision formats; see the Appendix G for details. *Stage2*: To prevent data leakage, we conduct RL training on the 10K prompts retained from the previous step, with GVM providing training feedback.

## 4.2 TRAINING SETTINGS

**Base setting.** We fine-tune a base model (*LLaMA3-8B-Base* and *LLaMA3.2-3B-Base*) to obtain an SFT model. The data are split into three disjoint parts: SFT, reward-model training, and RL. We then apply reinforcement learning to the SFT model. This setup follows the standard RL stage and is transparent and reproducible.

**Instruction setting.** We select an off-the-shelf instruction-tuned model as the SFT model. To facilitate comparison with related work(Li et al., 2023; Rafailov et al., 2024), we choose the *Mistral-7B-Instruct-v0.2* and *LlaMA3-8B-Instruct*. These models have undergone extensive instruction fine-tuning processes based on the base model, resulting in stronger capabilities compared to the SFT model in the Base setting. For more training details, please refer to Appendix C

## 4.3 BASELINE AND EVALUATION

**Baseline.** For GVM, we compare it with scalarRM trained on the same data. We also compare it with the value model in PPO training. In PPO, scalarRM provides the supervision signal. For the policy model, we compare it with the backbone model and with other RL algorithms, including PPO, GRPO, DPO, and ReMax. In all these baselines, scalarRM provides the supervision signal.

**Evaluation.** To evaluate GVM, we use a held-out test set and reward-related benchmarks (e.g., RewardBench). Because GVM produces token-level estimates for each response, we recover a response-level reward by applying a Bellman-style backup. We discuss our choice of evaluation metrics in the Appendix. We evaluate the response quality of the policy models using the most popular benchmarks: MT-Bench(Zheng et al., 2023), Arena-Hard(Li et al., 2024), and Alpaca-Eval(Dubois et al., 2024). Additionally, we compare the LLM-as-a-judge win rate on the held-out test set.

Table 1: Result of Base setting. Both methods were initialized from SFT and optimized. DVPO achieved the best benchmark scores across models with different parameter sizes. MT-Bench scores range from 0–10.

| Model | Llama3.2-3B-Base | | | Llama3-8B-Base | | |
|---|---|---|---|---|---|---|
| | Mtbench | Arena hard | AlpacaEval2 | Mtbench | Arena hard | AlpacaEval2 |
| SFT | 5.22 | 10.4 | 8.19 | 4.87 | 10.3 | 9.79 |
| PPO | 5.33 | 13.5 | 11.54 | 4.98 | 11.7 | 11.14 |
| Remax | 5.23 | 12.2 | 12.02 | 4.94 | 8.9 | 9.62 |
| GRPO | 5.46 | 13.4 | 10.86 | 4.91 | 10.4 | 10.27 |
| DVPO | **5.73** | **15.1** | **12.33** | **5.01** | **11.8** | **11.33** |
| Δ | +5.10% | +4.7% | +4.14% | +1.40% | +1.5% | +1.54% |

## 5 EXPERIMENT RESULTS

### 5.1 DVPO ON BASE SETTING

As mentioned earlier, to clearly evaluate the feasibility of DVPO compared to existing methods, we conducted experiments on publicly available datasets. The datasets were split proportionally for supervised fine-tuning (SFT) and reward learning (value learning). Subsequently, the resulting reward model (value model) was used to guide the policy optimization for preference learning. Given the relatively small amount of data at each stage (with the RL phase accounting for approximately 20% of the entire dataset), our focus in the Base setting is on relative improvement rather than absolute performance metrics. The results are presented in Table 1.

**DVPO demonstrates superior optimization performance.** In experiments conducted on LLaMA3-3B and LLaMA3-8B, DVPO consistently outperforms SFT significantly. Specifically, under the Base setting of LLaMA3-8B, DVPO achieves improvements of 0.2 on MT-Bench, 1.5 on Arena-Hard, and 1.74 on Alpaca-Eval compared to the SFT model. The improvements are even more pronounced with LLaMA3-3B, where DVPO achieves a 0.51 increase on MT-Bench, a 5-point increase on Arena-Hard, and a 4.14-point increase on Alpaca-Eval. Similarly, as shown in Figure 2, on the held-out test set, DVPO achieves a higher win rate compared to the SFT model. These results validate the robustness and effectiveness of DVPO.

**The pre-trained value model can provide a sufficiently high performance ceiling.** According to the experimental results in the Base setting, the final convergence performance of DVPO is very close to that of standard PPO (and in some cases, slightly exceeds it), indicating that the supervision signals provided by the pre-trained value model can support a sufficiently high performance ceiling. As an actor-critic method, PPO trains the policy model and value model simultaneously, with the reward model providing environmental feedback. The value model, as a bridge between immediate rewards (sentence-level) and long-term returns, offers finer-grained supervision signals (token-level) compared to immediate rewards alone. In DVPO, the global value model(GVM) is pre-trained on responses from different policies, enabling it to provide supervision feedback to various policies during the RL phase. The experimental results demonstrate that the fine-grained environmental feedback provided by DVPO can achieve a performance ceiling comparable to that of the actor-critic framework while significantly reducing training complexity and instability.

### 5.2 DVPO ON INSTRUCTION SETTING

To evaluate the effectiveness of DVPO in most RLHF scenarios (optimization starting from a pre-aligned model), we selected Mistral-7B-Instruct-v0.2 and LLaMA3-8B-Instruct as the SFT model. An existing UltraRM reward model was used to collect labeled data for training the value model, value model also initialized from SFT model. The trained value model then guided the preference optimization of the SFT model. The results are presented in Table 2. Experimental findings demonstrate that our method exhibits significant advantages over other approaches. Our main findings are as follows:

Table 2: Result of Instruction setting. **Instruction** refers to ***Mistral-7B-Instruct-v0.2*** and ***LlaMA3-8B-Instruct***. All methods are initialized from their respective instruction models. Results marked with an (*) are sourced from the original paper or inference from the open-source release.

| Method | Mistral 7B | | | Llama3 8B | | |
|---|---|---|---|---|---|---|
| | **Mtbench** | **Arena hard** | **AlpacaEval2** | **Mtbench** | **Arena hard** | **AlpacaEval2** |
| Instruction | 6.60* | 12.6* | 17.11* | 6.90* | 20.6* | 34.38* |
| ReMax | 6.67* | 21.9* | 20.55* | 7.62 | 30.8 | 36.04 |
| DPO | 6.30 | 16.3 | 26.80 | 7.01 | 32.6 | 40.32 |
| GRPO | 6.31 | 21.8 | 27.19 | 7.53 | 34.1 | 35.17 |
| PPO | 6.55 | 19.4 | 19.62 | 7.55 | 36.3 | 34.98 |
| DVPO | **6.79** | **24.7** | **27.43** | **7.72** | **39.2** | **42.59** |
| $\Delta$ | +1.90% | +12.1% | +10.32% | +8.20% | +18.6% | +8.21% |

**DVPO significantly enhances the performance of instruction models.** Compared to the original Mistral-7B-Instruct-v0.2, DVPO consistently demonstrates performance improvements across all benchmarks. Specifically, it achieves a 0.19 improvement on MT-Bench and increases the win rate by 12.1% on Arena-Hard. Additionally, it enhances the length-controlled win rate by 10.32% on Alpaca-Eval. Furthermore, compared to DPO, DVPO also exhibits notable advantages, on the Mistral setting, outperforming 8.4% in Arena-Hard and 0.49 in MT-Bench. On the Llama setting, outperforming 0.71 in MT-Bench. These results indicate that the pre-trained value model can effectively and reliably guide the optimization of policy models.

**Finer-grained feedback signals lead to superior performance.** Compared to reward-only methods such as ReMax and GRPO, DVPO demonstrates significant performance advantages on Mt-Bench, Arena-hard and Alpaca-Eval. In ReMax and GRPO, the feedback signal is at the sentence level from the reward model, assigning a single reward score to the entire sentence and treating all tokens as equally contributing to the overall score. This design prevents the policy model from learning fine-grained preferences during the RL phase, resulting in suboptimal performance. In contrast, the GVM—trained with TD-learning over prefixes—assigns different values to different parts of a response; as shown in Figure 3, decisive reasoning tokens receive high values while misleading continuations receive low values. By assigning a return value to each token, GVM facilitates more effective training of the policy model. In addition to providing fine-grained returns, DVPO retains the on-policy characteristics of the original PPO, enabling a larger exploration space and offering a higher performance ceiling.

## 6  ANALYSIS

### 6.1  COMPARSION OF SCALAR RM IN REWARD BENCH

To verify that GVM can effectively learn state-value estimates from offline trajectories, we aggregate its token-level values into a final, outcome-level reward and evaluate on reward-related benchmarks. We also compare against scalar reward models (ScalarRM) trained on the same data. As shown in Table 3, the average RewardBench (Lambert et al., 2024) scores of GVM and ScalarRM are comparable overall, but their performance profiles differ. On the Chat (easy) subset, ScalarRM consistently outperforms GVM, suggesting that sequence-level prediction with a pairwise Bradley–Terry (BT) loss readily captures global objectives and models overall preference on easier cases. In contrast, on the Chat-Hard subsets, GVM—trained with Bellman/temporal-difference (TD) learning for token-level value estimation—generalizes better to more complex problems. In sum, given identical data, GVM and ScalarRM achieve similar aggregate performance, but GVM provides finer-grained and more accurate supervision at the token level; relative to distributing a single sequence-level reward down to tokens, these direct value signals improve policy learning. Additionally, we provide illustrative GVM case studies in Appendix D.

### 6.2  COMPARSION OF ACTOR-CRITIC VALUE MODEL

Beyond our comparison with ScalarRM, we also analyze the value model in the actor–critic paradigm. As shown in Table 4, on in-distribution data GVM significantly outperforms the A/C value model,

Table 3: In the RewardBench results, GVM and ScalarRM are trained on the same data.

| Model | Method | Chat | Chat-Hard | Safety | Reason. |
|-------|--------|------|-----------|--------|---------|
| Llama3 8B | GVM | 88.3 | **67.5** | 66.8 | **79.2** |
| | SclarRM | **95.5** | 58.5 | 73.9 | 65.4 |
| Mistral 7B | GVM | 75.4 | **61.4** | 66.7 | **64.4** |
| | SclarRM | **93.8** | 52.4 | 64.1 | 60.3 |

Table 4: Result of Ultrafeedback testset. A/C denotes the PPO critic (value model), evaluated at PPO training convergence.

| Method | Llama3 8B | Mistral 7B |
|--------|-----------|------------|
| GVM | **68.1** | **64.5** |
| A/C | 60.6 | 57.6 |

Table 6: Memory footprint and computational cost analysis. $M_{\text{train}}$: training-time memory (parameters, optimizer states, activations); $M_{\text{infer}}$: inference-time memory; $C_{\text{gene}}$: computational cost for generating $n$ responses; $C_{\text{back}}$: computational cost for backpropagation and optimizer update.

| | **ScalarRM** | **GVM** | **PPO** | **ReMax** | **GRPO** | **DVPO** |
|---|---|---|---|---|---|---|
| $M_{\text{train}}$ | $m_{\text{train}}$ | $m_{\text{train}}$ | $2 \times m_{\text{train}}$ | $m_{\text{train}}$ | $m_{\text{train}}$ | $m_{\text{train}}$ |
| $m_{\text{infer}}$ | $-$ | $-$ | $2 \times m_{\text{infer}}$ | $2 \times m_{\text{infer}}$ | $2 \times m_{\text{infer}}$ | $2 \times m_{\text{infer}}$ |
| $C_{\text{gene}}$ | $-$ | $-$ | $c_{\text{gene}}$ | $2 \times c_{\text{gene}}$ | $n \times c_{\text{gene}}$ | $c_{\text{gene}}$ |
| $C_{\text{back}}$ | $c_{\text{back}}$ | $c_{\text{back}}$ | $2 \times c_{\text{back}}$ | $c_{\text{back}}$ | $c_{\text{back}}$ | $c_{\text{back}}$ |

indicating that a value model pretrained directly from preference data more effectively estimates response state values than the critic jointly trained with PPO. Moreover, this advantage transfers across different base (backbone) models. A/C value model is trained to fit returns under the current policy. It is tightly coupled to the policy, and its data distribution shifts during training, so it best predicts returns for outputs likely under that policy. In contrast, GVM learns from large-scale preferences which state transitions increase or decrease reward. It is policy-agnostic and therefore yields more accurate value estimates.In this sense, our value model is "global" in that (i) it is policy-agnostic rather than coupled to a single behavior policy, and (ii) it evaluates each action from a trajectory-level perspective by considering how the entire trajectory contributes to the final outcome.

To further compare the performance of GVM and A/C under distribution shifts, we evaluate the GVM and A/C value model from Ultrafeedback experiment on a new distribution, new prompts in the test split of the HH-RLHF dataset. The result presented in Tabel 5, We also compute the accuracy metrics. On the new distribution, both GVM and A/C show lower performance, but GVM still leads. This indicates that GVM's pretrained value estimates remain robust under shift. Pretraining on offline trajectories helps the value model learn value estimation more faithfully and improves generalization.

Table 5: Result of HH-RLHF testset.

| Method | Llama3 8B | Mistral 7B |
|--------|-----------|------------|
| GVM | **63.3** | **60.8** |
| A/C | 57.5 | 53.8 |

## 6.3 Comparison of Computation

As the Table 6 shows, training GVM and ScalarRM requires nearly the same GPU memory. Both use the same base model with a single linear head (hidden size $\rightarrow 1$). Their training time is also similar, since each step performs one backpropagation. Overall, the compute budgets for GVM and ScalarRM are essentially equivalent. Among RL methods, PPO is the most resource-intensive because it trains two models in parallel (policy and value). ReMax, GRPO, and DVPO have comparable memory costs, but group-based methods such as GRPO must generate multiple responses per prompt, which increases training time. In sum, DVPO offers the best balance of memory and time and, with token-level supervision from GVM, outperforms the other methods. We provide a detailed analysis of computational resources and the training curves in the Appendix F.

## 7 Discussion

**Conclusion.** We propose Decoupled Value Policy Optimization (DVPO), a framework that eliminates joint actor-critic training in RLHF by leveraging a pretrained global value model (GVM). Our theoretical analysis proves the functional equivalence of reward and value models under the constraint

of no new reward feedback, justifying the use of a fixed GVM for efficient and scalable optimization. Empirical results demonstrate that DVPO achieves comparable or surpasses performance to state-of-the-art RLHF methods on multiple benchmarks. Future work will focus on refining the value model's training process to enhance estimation performance.

**Limitations.** DVPO assumes that the offline preference dataset provides adequate coverage of relevant trajectories and that the Global Value Model (GVM) can approximate token-level returns within bounded error. However, the need for sufficiently diverse preference data is inherent to reward learning, not specific to the GVM. Unlike joint actor–critic training, DVPO does not adapt the critic as the policy evolves; thus its guidance is fixed and may fail in highly non-stationary settings. Our equivalence analysis also holds only under the constraint of no new reward signals—if additional human or environment feedback becomes available during training, a static GVM cannot fully exploit it. A promising future work to mitigate this limitation is to extend DVPO to a semi-online regime by periodically refreshing the GVM with newly collected preference data, while carefully controlling the update frequency to preserve the stability of the training dynamics. Finally, although DVPO reduces computation and improves stability, it may inherit biases from the reward-preference data itself, which remain an open challenge for all RLHF methods.

## ETHICS STATEMENT

We adhere to the ICLR Code of Ethics. This study did not use personally identifiable information or sensitive attributes, and involved no direct interaction with human subjects. Our institution determined that the work does not constitute human-subjects research (or qualifies for exemption); therefore, IRB approval was not required. There are no competing interests or sponsorships that could unduly influence the results.

## REPRODUCIBILITY STATEMENT

Our study uses open-source datasets; their names and download URLs are listed in the relevant sections of the paper. Our complete training and evaluation code is provided in `https://github.com/microsoft/DKI_LLM/tree/main/dvpo`. The appendix details the full hyperparameter settings and dataset splits. Where permitted, we release model checkpoints to facilitate verification of the reported results. A permanent public repository will be made available upon acceptance.

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

# A    USE OF LARGE LANGUAGE MODELS (LLMS)

We used a large language model solely for sentence-level copyediting (grammar, clarity, and fluency) of text written by the authors. All LLM suggestions were reviewed and accepted by the authors, who take full responsibility for the final content.

# B    PROOF

*Proof.* **Converting $R_\phi$ to a Value Function.** Using the pretrained reward model $R_\phi(s, a)$, we compute a surrogate action-value function $\widetilde{Q}_\phi^R(s, a)$ by solving the Bellman equation offline:

$$\widetilde{Q}_\phi^R(s, a) = R_\phi(s, a) + \gamma \, \mathbb{E}_{s' \sim P}\Big[\mathbb{E}_{a' \sim \pi_\theta}[\, \widetilde{Q}_\phi^R(s', a')]\Big].$$

Under standard offline RL assumptions (including sufficient coverage of the state-action space), $\widetilde{Q}_\phi^R(s, a)$ converges to an approximation of $Q^\pi(s, a)$ with an additional error term $\epsilon_{R2Q}$ that depends on $\epsilon_R$.

**Converting $Q_\psi$ to an Approximate Reward Signal.** Conversely, if $Q_\psi(s, a)$ approximates $Q^\pi(s, a)$, then rearranging the Bellman equation yields an approximate reward:

$$\widetilde{R}_\psi(s, a) = Q_\psi(s, a) - \gamma \, \mathbb{E}_{s' \sim P, \, a' \sim \pi_\theta} \left[Q_\psi(s', a')\right].$$

The error in this reconstruction is bounded by $\epsilon_{V2R}$, which is a function of $\epsilon_Q$.

**Policy Gradient Equivalence.** We now explicitly define the two policy gradient objectives used in our analysis.

The policy gradient with a reward-derived value function, denoted as $\nabla_\theta \mathcal{J}_R(\pi_\theta)$, is given by:

$$\nabla_\theta \mathcal{J}_R(\pi_\theta) = \mathbb{E}_{s \sim d^{\pi_\theta}, \, a \sim \pi_\theta} \left[\nabla_\theta \log \pi_\theta(a \mid s) \cdot A_R(s, a)\right],$$

where the advantage is computed using a surrogate value function $\widetilde{Q}_\phi^R(s, a)$ obtained by solving the Bellman equation using the pretrained reward model $R_\phi(s, a)$:

$$A_R(s, a) = \widetilde{Q}_\phi^R(s, a) - b(s).$$

The policy gradient using a pretrained global value model, denoted as $\nabla_\theta \mathcal{J}_Q(\pi_\theta)$, is defined similarly:

$$\nabla_\theta \mathcal{J}_Q(\pi_\theta) = \mathbb{E}_{s \sim d^{\pi_\theta}, \, a \sim \pi_\theta} \left[\nabla_\theta \log \pi_\theta(a \mid s) \cdot A_Q(s, a)\right],$$

where the advantage is:

$$A_Q(s, a) = Q_\psi(s, a) - b(s),$$

and $Q_\psi$ is the pretrained global value model.

In both cases, $b(s)$ is a baseline (e.g., state value or Monte Carlo average) used to reduce variance. This aligns with standard actor-critic or REINFORCE-style policy gradient frameworks (Sutton, 2018). The gradients are computed using samples from the current policy $\pi_\theta$, and the fixed models $R_\phi$ and $Q_\psi$ provide the learning signals.

We assume:

$$\left|\widetilde{Q}_\phi^R(s, a) - Q^\pi(s, a)\right| \le \epsilon_{R2Q}, \quad \left|Q_\psi(s, a) - Q^\pi(s, a)\right| \le \epsilon_Q,$$

and thus the difference in advantage estimates is bounded:

$$\|A_R(s, a) - A_Q(s, a)\| \le \epsilon_{R2Q} + \epsilon_Q.$$

Combining the expressions above, the policy gradients differ by:

$$\|\nabla_\theta \mathcal{J}_R(\pi_\theta) - \nabla_\theta \mathcal{J}_Q(\pi_\theta)\| \le \kappa(\epsilon_R, \epsilon_Q),$$

where $\kappa$ is a bounded, non-negative function of the approximation errors, satisfying $\kappa(\epsilon_R, \epsilon_Q) \to 0$ as $\epsilon_R, \epsilon_Q \to 0$.

This formalizes our claim that, under no new reward feedback, the reward-derived and value-derived gradients converge to the same update direction.

**Static Supervision.** Since no additional ground-truth rewards are collected, both $R_\phi$ and $Q_\psi$ remain fixed throughout policy training. Consequently, their induced gradients remain constant over time, ensuring that the policy updates they drive are equivalent up to the bounded error $\kappa(\epsilon_R, \epsilon_Q)$.

The steps above demonstrate that, under our assumptions, the policy gradient updates based on the pretrained reward model and the global value model are equivalent as the approximation errors vanish. This completes the proof. $\square$

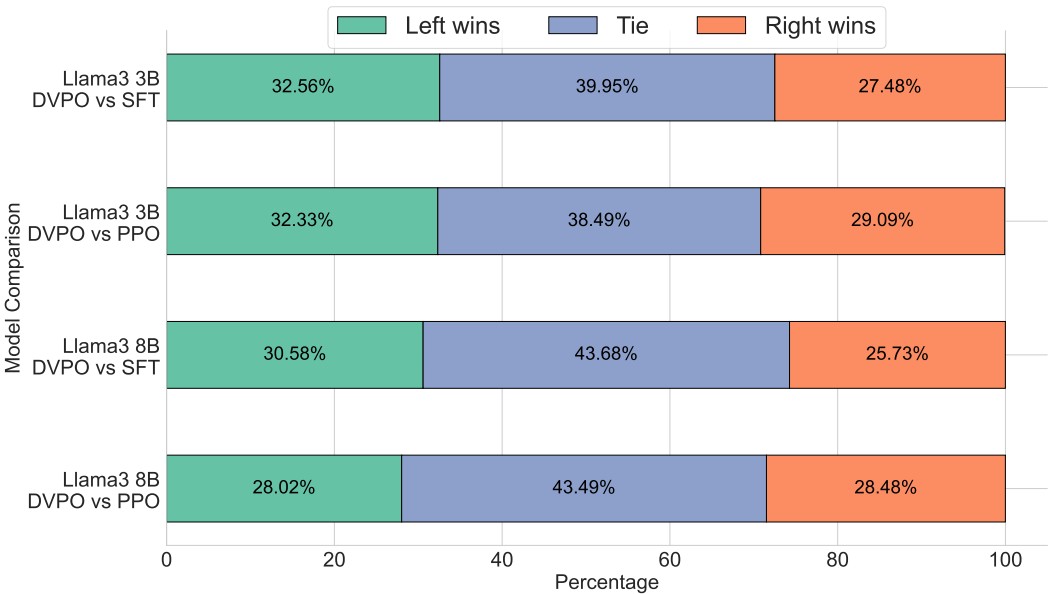

Figure 2: Results of the model on the Ultrafeedback held-out testset. We employed GPT4o as a judge to assess the quality of model-generated responses. Performance is measured using the win rate, where **Left** represents DVPO, and Right represents the baseline model for comparison.

## C  TRAINING DETAILS

**SFT training**. We use the following hyperparameters for instruction fine-tuning training. We employ a learning rate of 2e-5 with cosine decay, 2 warmup steps, and a batch size of 16. We calculate the loss only for the target tokens rather than the full input sequence, and we train for 3 epochs on the training data. we conduct the training on 8 NVIDIA A100 80G GPUs.

**Reward training**. To enable the model to learn the relative ranking among different responses, we use a pair-wise loss. The task type is sequence classification. We use a learning rate of 2e-5 with linear decay and the AdamW optimizer for training over 2 epochs, with a batch size of 4. We conduct the training on 8 NVIDIA A100 80G GPUs .

**DVPO and Baselines training**. For PPO training, we use a learning rate of 2e-6 and set the maximum generated sequence length to 1024. We employ a batch size of 8 and a mini-batch size of 2, with 4 PPO epochs and 2 gradient accumulation steps. The KL coefficient is set to 0.05, and in Appendix H we conducted an ablation study on the KL coefficient. The target KL divergence $\text{KL}_{\text{target}}$ is set to 0.1. For a fair comparison, training for DVPO and other baselines was conducted using the same hyperparameter settings (e.g., batch size, sequence length, etc.). For GRPO training, we roll out 5 responses for each prompt.

**Global value model training.** We initialize the value model from the SFT checkpoint and optimize a token-level mean-squared error objective with a discount factor $\gamma = 1$. The training is conducted with a batch size of 32, a sequence length of 1024, and a learning rate of 2e-6.

Figure 3: An example of the supervisory signal provided by a Global Value Model (**GVM**). The GVM is capable of providing token-level feedback. In this example, the GVM assigns a **lower value** to the incorrect response (**response2**: *"is an island"*) and a **higher value** to the critical token *"not"* in the correct response (**response1**: *"not an island"*).

## D    GVM CASE STUDY

For the same question, "As an island, is Beijing located in Asia?", the value model provides fine-grained supervisory signals for two different responses. The GVM assigns specific values to each token in the responses. These values represent the model's assessment of the importance or correctness of each token in the given context.

As shown in Figure 3. For Response 1, the critical token "not" is given a higher value (0.2099), highlighting its significance in forming the correct response, "not an island." For Response 2, the GVM assigns lower values to incorrect tokens, such as "is" (-0.6177) and "an" (-0.4766), indicating their contribution to the incorrect response, "is an island." This token-level evaluation demonstrates the GVM's ability to guide learning by penalizing incorrect responses and reinforcing critical tokens in correct responses, thereby enhancing training accuracy and interpretability.

## E    GVM PERFORMANCE

We evaluate the performance of the GVM from multiple perspectives. Specifically, we observe that the GVM assigns higher value scores to good actions and lower value scores to bad actions, as illustrated in Figure 3. We evaluate the global value model(GVM) on a held-out test set. For each question, we expect the value model to assign higher values to good responses compared to bad ones. We calculate the accuracy under various metrics (mean, P1, etc.) to assess the model's performance.

Table 8 presents the performance metrics of the Global Value Model (GVM) initialized from different size models. The metrics include the mean value, which indicates the average score assigned by the GVM, and various percentiles (P1, P5, P10, P90, P95, and P99), representing the corresponding value distributions. The accuracy metric evaluates the proportion of cases where the GVM correctly assigns a higher value to a good response compared to a bad response, thereby reflecting the effectiveness of the GVM in providing accurate feedback.

As shown in the table, the value model's performance exhibits a clear scaling law, with larger models achieving higher accuracy. Moreover, when trained on the same dataset, it delivers superior optimization outcomes compared to the reward model. These results demonstrate the GVM's ability to provide reliable token-level evaluations.

## F    ANALYSIS OF COMPUTATION EFFICIENCY

DVPO improves upon RLHF by leveraging a pre-trained value model to provide environmental feedback, resulting in enhanced computational efficiency, stability, and convergence speed.

**Memory Efficiency.** As shown in Table 7, the experimental setup for standard PPO and DVPO was kept identical. Under standard training conditions, DVPO achieves approximately a 30-40%

Table 7: Computational results under the **Base setting**. For the 3B model, we used Deepspeed **Zero-1** strategy; for the 8B model, due to OOM issues with Zero-1, we used **Zero-3**. We report GPU memory consumption per GPU (GB) and time per step (s) on an 8 × A100 setup.

| Method | Llama3-8B-Base | | Llama3.2-3B-Base | |
|--------|----------------|----------------|------------------|----------------|
| | Mem (GB) | Time (s) | Mem (GB) | Time (s) |
| PPO | 78.96 | 70.24 | 41.43 | 23.00 |
| ReMax | 61.23 | 65.96 | 26.87 | 44.13 |
| GRPO | 64.35 | 57.93 | 28.20 | 22.17 |
| DVPO | 60.49 (-23.4%) | 48.00 (-31.7%) | 27.50 (-33.6%) | 13.00 (-43.5%) |

reduction in memory usage. For instance, with LLaMA-3B, we trained using Zero-1 on 8 A100 GPUs with a batch size of 4. Standard PPO requires 41.43 GB of memory due to the need to simultaneously load 4 models (policy model, critic model, reference model, and reward model), with two models (policy model and critic model) requiring activation. In contrast, DVPO only requires 27.5 GB of memory, as it only loads the policy model, global value model, and reference model, with only the policy model being activated during training. For LLaMA-8B, we used DeepSpeed Zero-3 with a batch size of 4. Even under this configuration, DVPO effectively reduces memory usage, enabling support for larger batch sizes during training.

**Time Efficiency.** DVPO accelerates the training process, requiring only half the time per batch compared to standard PPO under the same batch size. This improvement stems from the fact that, during the generation phase, DVPO requires outputs from only three models, whereas standard PPO relies on four. Additionally, during the backpropagation phase, PPO updates both the policy and value models, while DVPO updates only the policy model. Overall, under identical settings (batch size), DVPO achieves a 1.76x speedup in training ($1.76 \approx 23/13$). By employing the GVM for advantage estimation, DVPO requires fewer generated samples during the rollout phase and thus achieves faster training compared to other memory-efficient RL algorithms (e.g., ReMax and GRPO).

**Training Step.** We observe that using a pre-trained value model to provide token-level fine-grained environmental feedback enables faster model convergence, which aligns with the findings in (Noukhovitch et al., 2024). Specifically, under identical experimental settings (same batch size, experience replay count, etc.), DVPO requires only approximately $3/4$ of the interaction steps to converge. As shown in Figure 4, in the LLaMA3-8B experiment, the reward curve of PPO reaches its peak at 1250 steps, whereas DVPO achieves optimal performance around 800 steps. Similarly, in LLaMA3-3B, PPO reaches its peak at 600 steps, while DVPO converges in 350 steps. Since DVPO provides token-level supervision feedback and is pre-trained in advance, it can be considered a form of warm-starting, thereby accelerating policy convergence.

## G   ANALYSIS OF REWARDS ASSIGNED

We compared two distinct reward distribution schemes, and the resulting training performance is shown in the Table 9.

**From Reward Model** For scenarios in which a reward model is available (used for a fair comparison to PPO), we first leverage preference data (e.g., tuples of the form (Q, $A_1$,$A_2$)) to obtain scalar scores $S_1$ and $S_2$ for $A_1$ and $A_2$ via the reward model. Each tuple (Q, A, S) is then treated as a trajectory, with S serving as the scalar return, and the GVM is trained on this token-level data using temporal-difference learning.

**From Preference Data.** For scenarios without a reward model (for broader applicability), we learn the value model directly from preference data. Specifically, we assign +1 to the preferred response and −1 to the less preferred one; these labels are treated as scalar returns to supervise the GVM directly, obviating the need for any reward model.

With the same data and base model, the value model from reward model accuracy is 0.64, and the value model from preference data is 0.67. The two training approaches produce virtually identical outcomes, demonstrating the GVM's tolerance for disparate data formats and its ability to be trained for varying application scenarios.

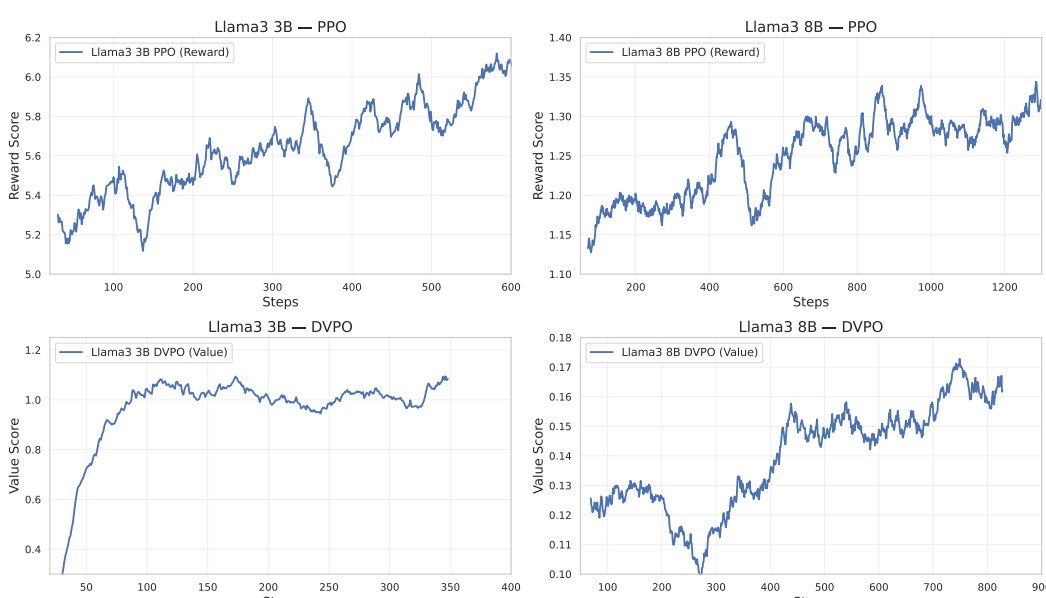

Figure 4: Learning curve of the policy model during the RL stage under the Base setting. DVPO demonstrates faster and more stable convergence compared to other methods.

Table 8: Value model performance in different size models.**Mean Value** is the mean of the value scores over the trajectory. **P**$k$ **Value** denotes the $k$-th percentile of the value distribution in trajectory.

| METRIC | LLAMA3 8B | LLAMA2 13B | LLAMA3 3B | MISTRAL 7B |
|---|---|---|---|---|
| MEAN VALUE | 68.05 | 70.55 | 63.61 | 64.51 |
| P1 VALUE | 57.15 | 63.80 | 55.00 | 56.02 |
| P5 VALUE | 60.24 | 65.90 | 57.15 | 58.46 |
| P10 VALUE | 62.50 | 67.70 | 58.45 | 59.83 |
| P90 VALUE | 66.23 | 69.05 | 64.40 | 61.47 |
| P95 VALUE | 67.95 | 67.55 | 62.55 | 60.30 |
| P99 VALUE | 66.3 | 64.75 | 56.99 | 59.94 |

# H   ABLATION ON THE KL COEFFICIENT

To search the suitable KL coefficient, we performing a grid search in the range $\{0.01, 0.05, 0.10\}$. For each setting, we evaluate the resulting policy on MT-Bench, Arena-Hard, and AlpacaEval2. As shown in Table 10, DVPO achieves consistently strong performance across all three values, and remains superior to the baseline under every tested KL coefficient. We adopt $0.05$ as the default value in our main experiments, which is aligned with common practice in prior RLHF work.

# I   GPT4 EVALUATION PROMPT

A crucial element of our experimental framework is the evaluation of win rates using GPT-4. In this section, we provide the prompts utilized to generate win rates for held-out test set in Ultrafeedback dataset. All experiments were conducted using the gpt-4o-20240806 model. The sequence of responses was randomized for each evaluation to ensure unbiased results.

**GPT-4 as judge system prompt:**

Review the user's question and the corresponding response using the additive 5-pointscoring system described below. Points are accumulated based on the satisfaction of each criterion:

Table 9: Comparison of learning value models from reward data and preference data

| ACCURACY | FROM REWARD | FROM PREFERENCE |
|---|---|---|
| MEAN VALUE | 64.51 | 67.85 |
| P1 VALUE | 56.02 | 58.60 |
| P5 VALUE | 58.46 | 62.80 |
| P10 VALUE | 59.83 | 63.95 |
| P90 VALUE | 61.47 | 65.85 |
| P95 VALUE | 60.30 | 65.80 |
| P99 VALUE | 59.94 | 64.05 |

Table 10: Ablation on the KL coefficient for DVPO.

| KL | MT-Bench | Arena-Hard | AlpacaEval |
|---|---|---|---|
| 0.01 | **8.50** | 37.50 | 41.66 |
| 0.05 | 7.72 | **39.20** | **42.59** |
| 0.10 | 7.73 | 38.60 | 40.96 |

- Add 1 point if the response is relevant and provides some information related to the user's inquiry, even if it is incomplete or contains some irrelevant content.

- Add another point if the response addresses a substantial portion of the user's question, but does not completely resolve the query or provide a direct answer.

- Award a third point if the response answers the basic elements of the user's question in a useful way, regardless of whether it seems to have been written by an Al Assistant or if it has elements typically found in blogs or search results.

- Grant a fourth point if the response is clearly written from an Al Assistant's perspective, addressing the user's question directly and comprehensively, and is well-organized and helpful, even if there is slight room for improvement in clarity, conciseness or focus.

- Bestow a fifth point for a response that is impeccably tailored to the user's question by an AI Assistant, without extraneous information, refecting expert knowledge, and demonstrating a high-quality, engaging, and insightful answer.

After examining the user's instruction and the response, please first give the total score. Then provide a brief explanation of your total score, up to 100 words. Output with the following format: Score: <total score> Evaluation evidence: <your brief explanation here>

Remember to assess from the Al Assistant perspective, utilizing web search knowledge as necessary. To evaluate the response in alignment with this additive scoring model, we'll systematically attribute points based on the outlined criteria.

**User input prompt:**

```
<Question>:  {question}
<response>:  {answer}
```

Table 11: A case study for GPT4 evaluation: LLama3 8B SFT

| Input | Compare and contrast Harry Potter with Percy Jackson series. |
|---|---|
| **Llama3 8B SFT** | Sure, I'd be happy to help you compare and contrast the Harry Potter and Percy Jackson series!Both Harry Potter and Percy Jackson are popular fantasy series that have captured the hearts of readers around the world. Here are some key similarities and differences between the two:Similarities:1. Magical elements: Both series feature a magical aspect, with Hogwarts School of Witchcraft and Wizardry being the setting for Harry Potter and Camp Half-Blood for Percy Jackson.2. Mythological references: Both series draw heavily on Greek mythology, with Percy Jackson's stories being based on the Labors of Hercules and other Greek myths.3. Relatable protagonists: Both Harry and Percy are relatable and charming protagonists that readers can connect with.Differences:1. Setting: While both series take place in magical worlds, the settings are quite different. Harry Potter's stories take place in a more traditional school setting, while Percy Jackson's stories take place in a summer camp atmosphere.2. Tone: Harry Potter is a more lighthearted and comedic series, while Percy Jackson takes a darker and more serious tone, exploring themes of mortality and responsibility.3. Plot structure: Harry Potter has a more linear plot structure, with each book building on the previous one's events. Percy Jackson, on the other hand, has a more episodic structure, with each book focusing on a different adventure or challenge for the characters.4. Themes: While both series explore themes of good and evil, Harry Potter focuses more on the importance of friendship and loyalty, while Percy Jackson delves into the consequences of power and responsibility.In conclusion, while both Harry Potter and Percy Jackson share many similarities, their differences in setting, tone, plot structure, and themes offer unique reading experiences for readers. Both series are highly recommended for anyone looking for a magical and engaging read! |
| **GPT4-as-Judge** | Score: 3
Evaluation evidence: The response provides a relevant comparison of the Harry Potter and Percy Jackson series, highlighting key similarities and differences in magical elements, settings, tone, plot structure, and themes. It addresses the user's question by outlining basic elements and offering a useful comparison. However, the answer contains inaccuracies, such as misrepresenting the tone of both series and the nature of mythological references in Percy Jackson. Additionally, the response could be more precise and direct, as some points are not entirely accurate or comprehensive. Thus, it meets the criteria for a basic, useful response but lacks thoroughness and precision for a higher score. |

Table 12: A case study for GPT4 evaluation: LLama3 8B PPO

| Input | Compare and contrast Harry Potter with Percy Jackson series. |
|---|---|
| **Llama3 8B PPO** | Sure, I'd be happy to help you compare and contrast the Harry Potter and Percy Jackson series!Both Harry Potter and Percy Jackson are popular fantasy series that have captured the hearts of readers around the world. Here are some key similarities and differences between the two:Similarities:1. Magical elements: Both series feature a magical aspect, with Hogwarts School of Witchcraft and Wizardry being the setting for Harry Potter and Camp Half-Blood being the setting for Percy Jackson.2. Teenage protagonists: The main characters in both series are teenagers, navigating their way through adolescence while also facing epic challenges and learning valuable life lessons.3. Epic adventures: Both series involve the protagonists on thrilling quests, battling mythical creatures, and uncovering secrets that could change the course of history.4. Portal to another world: In both series, there is a gateway to another world (Harry's cupboard under the stairs and the portal at Camp Half-Blood) that allows the protagonists access to a realm filled with magical creatures and unparalleled adventure.Differences:1. Setting: While both series take place in magical worlds, the settings are quite different. Harry Potter is set in a medieval-style world with castles and magical creatures, while Percy Jackson is set in a modern-day world with technology and urban landscapes.2. Tone: The tone of the two series is quite different. Harry Potter is lighthearted and comedic, while Percy Jackson is darker and more serious, with themes of mortality and morality.3. Mythological references: While both series draw on mythological references, the specific myths and legends used are different. Harry Potter relies more on European and British myths, while Percy Jackson draws on Greek and Roman mythology.4. Magic systems: The magic systems in the two series are also different. Harry Potter uses a spell-based magic system, while Percy Jackson uses a more physical, athletic approach to magic.5. Protagonists: The protagonists in the two series are quite different. Harry Potter is a boy wizard who discovers his true identity and heritage, while Percy Jackson is a demigod who must navigate the complexities of his divine heritage and mortal life.6. Plots: The plots of the two series are distinct. Harry Potter focuses on the battle between good and evil, with Harry as the prophesized hero who must face off against Voldemort. Percy Jackson, on the other hand, is about the struggles of the demigods and the impact of their mortal lives on their divine heritage.In conclusion, both Harry Potter and Percy Jackson are captivating fantasy series that share some similarities, but also differ in many ways. While both offer engaging and entertaining reads, they appeal to different audiences and perspectives. |
| **GPT4-as-Judge** | Score: 3
Evaluation evidence: The response provides a relevant comparison of the Harry Potter and Percy Jackson series, addressing the user's question with a mix of similarities and differences. It covers key aspects such as magical elements, protagonists, and settings, offering a basic understanding of both series. However, the response contains inaccuracies, particularly in tone descriptions and settings, which detracts from its comprehensiveness and accuracy. While it offers a useful comparison, it does not fully address the user's question with the depth or precision expected from an AI Assistant, nor does it reflect expert knowledge or exceptional insight. |

Table 13: A case study for GPT4 evaluation: LLama3 8B DVPO

| Input | Compare and contrast Harry Potter with Percy Jackson series. |
|---|---|
| **Llama3 8B DVPO** | Sure, I'd be happy to help! Both the Harry Potter and Percy Jackson series are popular and successful book series that have captivated readers worldwide. Here are some key similarities and differences between the two:Similarities:1. Magical elements: Both series feature magical aspects, such as spells, wizards, and fantastical creatures.2. Young protagonists: The main characters in both series are young and face challenges and adventures that come with growing up in a magical world.3. Family themes: Both series emphasize the importance of family and the bonds between family members.4. Friendship: The series also highlight the importance of friendship and the power of having loyal and supportive friends.Differences:1. Genre: While both series can be classified as fantasy, Harry Potter is more traditionally fantasy-oriented, while Percy Jackson is urban fantasy with a contemporary setting.2. Setting: Harry Potter is set in a parallel universe where wizardry and magic are prevalent, while Percy Jackson is set in present-day America and Greece.3. Magic system: The magic systems in each series are different. In Harry Potter, magic is cast using wands and spells, while in Percy Jackson, magic is accessed through the use of magical coins and enchanted items.4. Gods and myths: Harry Potter is based on the idea of witches and wizards living among us, while Percy Jackson is based on Greek mythology and features the gods and goddesses of ancient Greece.5. Tone: Harry Potter is written in a more traditional, epic style, while Percy Jackson is written in a more contemporary, humorous tone.Ultimately, both series offer engaging, imaginative storytelling that captures the hearts and minds of readers. While they both explore similar themes, their unique settings, magic systems, and tones make them distinct from one another. I hope this helps! Let me know if you have any other questions. |
| **GPT4-as-Judge** | Score: 5
Evaluation evidence: The response thoroughly addresses the user's question by comparing and contrasting the Harry Potter and Percy Jackson series. It identifies key similarities, such as magical elements, young protagonists, and themes of family and friendship. The response also effectively highlights differences in genre, setting, magic systems, mythological influences, and tone. The information is well-organized, relevant, and engaging, demonstrating a comprehensive understanding of both series. The response is tailored specifically to the user's inquiry, providing insightful and expert knowledge without any extraneous information. |

