# OpenReview forum: "Pretrain Value, Not Reward: Decoupled Value Policy Optimization"
_ICLR.cc/2026/Conference — ICLR 2026 Poster_

### Official Review · Reviewer_sBiL · 2025-10-15

**Soundness:** 3
**Presentation:** 3
**Contribution:** 4
**Rating:** 8
**Confidence:** 3

**Summary:**

This paper presents Decoupled Value Policy Optimization, a framework for reinforcement learning from human feedback that replaces the typical online-learned critic or reward model with a single, offline-pretrained global value model. The central idea is that pretraining a value model on the same dataset used for reward modeling is informationally equivalent to the joint reward-critic learning process, but with advantages in stability and efficiency. DVPO demonstrates strong empirical results on standard RLHF benchmarks (MT-Bench, Arena-Hard, Alpaca-Eval), surpassing or matching state-of-the-art methods, while significantly reducing memory consumption and training time.

**Strengths:**

1. The paper is well-motivated and the equivalence between reward-then-critic pipelines and direct value pretraining is well-argued and convincingly formalized, highlighting under-recognized redundancy in the RLHF paradigm.

2. The framework is particularly interesting to the community as DVPO simplifies RLHF engineering by eliminating online critic training, reducing GPU memory use (40%) and time (35%), thus enabling larger models or faster iteration with fewer resources.

3. Experiments, ablations, and case studies clearly show that DVPO attains or improves upon baseline methods in terms of various metrics.

**Weaknesses:**

1. The framework assumes the offline preference or reward dataset is broad enough for effective generalization.

2. In scenarios where additional human or environmental feedback can be injected mid-training, DVPO might not benefit from it.

**Questions:**

1. What mechanisms exist to detect or mitigate GVM misgeneralization due to dataset bias or coverage gaps?

2. Can the model identify when its value predictions become unreliable or out-of-distribution?

---

> ### Author Response · Authors · 2025-11-22
> **Response to Weaknesses 1 & Weaknesses 2**
>
> ## Weakness 1: The framework assumes the offline preference or reward dataset is broad enough for effective generalization.
>
> **Response to Weakness 1**: We appreciate your insightful comment. We would like to clarify that our framework does **not** introduce any additional data requirements beyond what is already standard for reward–model–based RLHF. Training a reward model that generalizes well to in-domain  inherently requires a sufficiently broad and diverse offline preference dataset (i.e., prompts with varied chosen/rejected responses) to ensure robustness and to reduce reward hacking. Our GVM is trained on exactly the same data format as conventional reward models, and therefore relies on the *same* “broad-enough offline dataset” assumption rather than a stronger one.
>
> More broadly, the need for diverse preference data is a property of reward learning in general, not specific to GVM. In practice, reward models are routinely trained with responses generated under different sampling schemes (e.g., varying temperature/top-p/top-k) or even different policy models in order to obtain diverse chosen/rejected pairs and learn a stable signal. GVM shares this requirement but does **not** add extra complexity in terms of data collection or annotation. If anything, an explicit design goal of GVM is to *leverage* trajectories collected from multiple policies within a single global value function, which can mitigate sensitivity to any single policy. We will clarify this point in the revised manuscript and explicitly discuss this standard assumption in our limitations section.
>
> ## Weakness 2: In scenarios where additional human or environmental feedback can be injected mid-training, DVPO might not benefit from it.
>
> **Response to Weakness 2**: We thank your for raising this important scenario.  We agree that the ability to inject additional human or environmental feedback mid-training is a valuable direction, and we view it as a natural extension of our current framework.
>
> In our experiments, we adopt the *standard* “offline RLHF” / two-stage setup: a fixed preference (or reward) dataset is collected first, a value-like signal (reward model or GVM) is trained on it, and then frozen during policy optimization.       This is also how most existing reward-model–based RLHF pipelines are implemented in practice, where the reward model remains fixed within a given RL training run.
>
> However, our method does **not** preclude incorporating additional feedback.       Since GVM is trained on exactly the same type of data as a conventional reward model, any strategy that periodically refreshes the reward model with newly collected annotations (e.g., after the policy converges or at predefined checkpoints) can be directly applied to GVM as well: one can augment the offline preference dataset with new human/environmental feedback, update or fine-tune the GVM, and then continue DVPO training with the improved global value estimates.       In this sense, DVPO is compatible with iterative or semi-online RLHF loops;       our current work simply focuses on the widely used offline setting for clarity and cost efficiency.
>
> We will in the revised manuscript and add a discussion in the future-work section about extending DVPO with interleaved GVM updates when mid-training feedback is available. We would be very grateful for any suggestions you may have.

---

> > ### Author Response · Authors · 2025-11-22
> > **Response to Questions 1 & Questions 2**
> >
> > ## Question 1: What mechanisms exist to detect or mitigate GVM misgeneralization due to dataset bias or coverage gaps?
> >
> > **Response to Questions 1**: We thank the reviewer for this thoughtful question.  For GVM, the primary mechanisms we currently use focus on carefully evaluating its behavior both in-domain and under distribution shift.
> >
> > First, we monitor GVM’s performance on a held-out test split of the preference data (Table 8), which reflects its in-domain learning quality and can reveal overfitting or instability relative to the training set.  Second, we explicitly evaluate GVM under distribution shifts (Table 5), where the policy, prompt type, or dataset differs from the GVM training distribution;  degradation in these settings serves as a practical signal of bias or coverage gaps and thus of potential misgeneralization.
> >
> > Moreover, since GVM is trained on the same preference format as a standard reward model, it can be directly evaluated on existing reward-model benchmarks, which provides an additional way to assess cross-policy generalization.  Finally, beyond aggregate metrics, we also perform qualitative analysis of the value scores produced by GVM (Fig. 3), where human annotators (or a strong LLM) inspect whether the relative values assigned to different responses are reasonable.  Such case studies help surface localized misgeneralization patterns that may not be obvious from averaged metrics.
> >
> > When these diagnostics indicate systematic issues, one can adopt the same mitigation strategies commonly used for reward models (e.g., augmenting or rebalancing underrepresented prompt types).
> >
> > ## Questions 2: Can the model identify when its value predictions become unreliable or out-of-distribution?
> >
> > **Response to Questions 2**: We thank the reviewer for this insightful question.   In its current form, like most reward model, our framework does not include a separate, fully-fledged “uncertainty head” that explicitly flags each individual GVM prediction as unreliable or out-of-distribution.   Instead, we rely on a combination of (i) offline diagnostics and (ii) conservative regularization during DVPO training to limit the impact of potential misgeneralization.
> >
> > On the diagnostic side, as discussed in our response to Question 1, we evaluate GVM both on held-out in-domain test splits and on distribution-shifted settings (Tables 8 and 5).   These evaluations provide empirical evidence of where GVM’s value estimates remain reliable and where degradation starts to appear.   In addition, since GVM operates on the same preference format as standard reward models, it can be directly plugged into existing reward-model benchmarks to further probe cross-policy generalization and reveal potential coverage gaps.
> >
> > On the training side, DVPO follows standard RLHF safeguards such as a KL penalty to the reference policy and clipped advantages.   These mechanisms implicitly constrain the policy to stay close to regions where the training data is denser, which reduces the chance that the policy aggressively exploits spurious high-value estimates in poorly covered areas.

---

### Official Review · Reviewer_PVQp · 2025-10-19

**Soundness:** 2
**Presentation:** 3
**Contribution:** 1
**Rating:** 2
**Confidence:** 4

**Summary:**

This paper proposes a novel approach to improve PPO-based RLHF by avoiding explicit reward modeling and instead learning a human preference-consistent critic/value model. The authors transform the traditional two-stage RLHF pipeline into an offline critic training plus online value optimization framework, termed DVPO. Through theoretical analysis, the paper establishes an equivalence relationship between reward models and value models under fixed feedback conditions, providing theoretical foundations for training the Global Value Model (GVM) and deriving convergence guarantees for DVPO. Experiments are conducted on the UltraFeedback dataset using models ranging from 3B to 8B parameters, both with and without instruction tuning. Results demonstrate DVPO's effectiveness across settings on evaluation benchmarks including MT-Bench, AlpacaEval 2.0, and Arena-Hard.

**Strengths:**

1. **Clear presentation and theoretical rigor**: The paper is well-structured with clear insights in the introduction, particularly regarding the equivalence between training reward models and critic models based on fixed feedback. The visualization effectively depicts algorithmic differences between PPO and DVPO, and the theoretical analysis is well-organized.
2. **Novel algorithmic contribution**: The proposed alignment algorithm offers a compelling alternative to the conventional two-stage RLHF pipeline (reward modeling + preference optimization), contributing a simplified RLHF framework that effectively reduces variance in online critic learning and improves training stability.
3. **Comprehensive empirical evaluation**: The experimental studies demonstrate thoroughness in base model selection, baseline method comparisons, and evaluation benchmark choices.

**Weaknesses:**

1. **Unclear mechanism of performance gains**: The underlying reasons for DVPO outperforming baselines remain insufficiently explained. The value model trained on pre-collected preference feedback effectively learns $V^{\pi_B}$ (assuming responses are collected from some behavior policy $\pi_B$), which contradicts the paper's claim that this value model is a "global" one (definitely not $V^*$). Additionally, the paper attributes superiority over sentence-level reward-based methods to GVM providing more fine-grained feedback. However, since human preferences are provided at the sentence level, the results and analysis do not adequately support this claim.
2. **Non-standard hyperparameter settings**: The experimental setup uses a batch size of 8, whereas current standard practice in the field uses batch sizes of 32 to 128. The KL coefficient is crucial for algorithm performance in both DVPO and baseline methods (including PPO and GRPO), and providing the search range would enhance experimental credibility.
3. **Insufficient justification for explicit value modeling**: The significance of explicitly fitting a value model needs stronger empirical backing. Given that GRPO and RLOO use Monte Carlo-based methods to estimate values, the authors should provide deeper insights into how separately training a value model enables DVPO to outperform these baselines. Furthermore, the comparison in Table 6 is unfair to GRPO regarding response generation time—Monte Carlo-based methods use an n-shot sampling strategy, making generation time equivalent to DVPO given the same training batch size.

**Questions:**

1. Have you considered incorporating Bradley-Terry reward modeling in the pre-training of value models? This is the most standard approach to reward modeling, as human feedback typically comes in pairwise comparisons. Simply assigning -1 and +1 rewards does not reflect the actual reward modeling procedure.

---

> ### Author Response · Authors · 2025-11-22
> **Response to Weaknesses1 (1/2)**
>
> ## Comment 1.1: The underlying reasons for DVPO outperforming baselines remain insufficiently explained.
>
> **Response to Comment1.1** :  We thank the reviewer for raising this point. DVPO outperforms existing RLHF baselines for two main reasons. First, DVPO avoids the well-documented instability of jointly training a policy and value model in PPO-style actor–critic RL for LLMs. Prior work has repeatedly shown that online actor–critic training in language-model RL suffers from critic drift, noisy value estimates, and optimization instability ([3][4] ). DVPO addresses this issue by pretraining the Global Value Model (GVM) offline using multi-policy trajectories and keeping it frozen during policy optimization. This decoupled design provides a significantly stronger and more stable value estimator. As shown in Tables 4 and 5, the pretrained GVM outperforms PPO critics both in-distribution and out-of-distribution, and DVPO built on top of it achieves the best overall policy quality.
>
> Second, DVPO provides token-level advantages through TD-learning over the GVM, yielding more accurate credit assignment. Sequence-level Monte-Carlo baselines such as GRPO and RLOO assign a single scalar advantage to all tokens in a response, which prevents them from distinguishing decisive, harmful, or redundant parts of a generation. In contrast, DVPO applies TD-learning over the GVM, training it to predict the expected final preference from every prefix of the response. TD propagation distributes sentence-level preferences across these prefixes, producing distinct token-level Q estimates that reflect each token’s contribution to the eventual outcome. This fine-grained, prefix-based decomposition produces substantially more informative gradients. Prior work has also observed that finer-grained credit assignment improves RLHF optimization effectiveness ([1][2]), and our results are consistent with this trend.
>
> Table 4: Result of Ultrafedback testset. A/C denotes the PPO critic (value model), evaluated at PPO training convergence.
>
> | Method | Llama3 8B | Mistral 7B |
> |--------|-----------|------------|
> | GVM    | **68.1**  | **64.5**   |
> | A/C    | 60.6      | 57.6       |
>
>
> Table 5: Result of HH-RLHF testset.
>
> | Method | Llama3 8B | Mistral 7B |
> |--------|-----------|------------|
> | GVM    | **63.3**  | **60.8**   |
> | A/C    | 57.5      | 53.8       |
>
> [1]Zeng, Y., Liu, G., Ma, W., Yang, N., Zhang, H., & Wang, J. (2024). Token-level direct preference optimization. arXiv preprint arXiv:2404.11999.
>
> [2]Ouyang, Y., Wang, L., Yang, F., Zhao, P., Huang, C., Liu, J., ... & Sun, F. (2025, November). Token-level proximal policy optimization for query generation. In Proceedings of the 2025 Conference on Empirical Methods in Natural Language Processing (pp. 31184-31198).
>
> [3]Shengyi Huang, Tianlin Liu, and Leandro Von Werra. The n implementation details of rlhf with ppo. In The Third Blogpost Track at ICLR 2024, 2024
>
> [4]Shao, Z., Wang, P., Zhu, Q., Xu, R., Song, J., Bi, X., ... & Guo, D. (2024). Deepseekmath: Pushing the limits of mathematical reasoning in open language models. arXiv preprint arXiv:2402.03300.
>
> ## Comment 1.2. The value model trained on pre-collected preference feedback effectively learns $V^{\pi_B}$ (assuming responses are collected from some behavior policy $\pi_B$), which contradicts the paper's claim that this value model is a "global" one (definitely not $V^*$).
> **Response to Comment 1.2**: We thank the reviewer for pointing this out. The “global” refers to two properties of our value model: (i) it is trained on trajectories generated by a *diverse mixture* of policies, rather than on-policy rollouts from a single behavior policy; and (ii) it is trained to look at the *entire trajectory* and estimate how the current action contributes to the final outcome, thus evaluating each action from a trajectory-level perspective.
>
> This “global” nature is also supported empirically. Table 3 shows that the global value model matches or improves over the scalar reward model on reward-bench; Table 4 and 5 show that it outperforms actor–critic value models both in-distribution and out-of-distribution; and Table 1 and 2 demonstrate that, when used in DVPO, it leads to stronger policy performance than the considered baselines. These results indicate that the global value model provides a more effective and broadly useful value estimate than both scalar RMs and actor–critic value models, which is precisely what we intend to capture with the term “global.”
>
> table 3: GVM vs ScalarRM in reward-bench
> | Model      | Method   | Chat | Chat-Hard | Safety | Reason. | Avg.   |
> |------------|----------|------|-----------|--------|---------|--------|
> | Llama3 8B  | GVM      | 88.3 | 67.5      | 66.8   | 79.2    | **75.45** |
> |            | ScalarRM | 95.5 | 58.5      | 73.9   | 65.4    | 73.33 |
> | Mistral 7B | GVM      | 75.4 | 61.4      | 66.7   | 64.4    | 66.98 |
> |            | ScalarRM | 93.8 | 52.4      | 64.1   | 60.3    | **67.65** |

---

> ### Author Response · Authors · 2025-11-22
> **Response to Weaknesses1 (2/2)**
>
> ## Comment 1.3: Additionally, the paper attributes superiority over sentence-level reward-based methods to GVM providing more fine-grained feedback. However, since human preferences are provided at the sentence level, the results and analysis do not adequately support this claim.
>
> **Response Comment 1.3**: Thank you for the question. Although human supervision is provided only at the sentence level, the GVM still obtains token-level credit through temporal-difference (TD) learning, which defines a value for every prefix of the response. For each training example, the final human preference score is propagated across all prefixes (e.g., “Paris”, “Paris is”, “Paris is not”, …), and the model is trained to predict the expected outcome from each prefix. Because different prefixes contain different semantic evidence about the final correctness, their ground-truth values are necessarily different. Thus, even with sentence-level supervision, TD learning yields different Q values for different tokens, enabling fine-grained feedback.
>
> Sentence-level reward-based methods cannot recover this structure: they assign the same scalar reward to every token in the sequence and therefore cannot distinguish which token made the answer correct or incorrect. For example, in the correct answer “Paris is not a country”, the prefix “Paris is not” already strongly indicates the correct final conclusion, whereas the prefix “Paris is” does not. To minimize prediction error, the GVM must assign a higher value to the token “not,” while reward-only methods give “not” exactly the same feedback as every other token. Figure 3 in our paper also provides a case study of GVM.
> This is why GVM provides genuinely finer-grained supervision and leads to more effective policy optimization, even though all methods use the same sentence-level human preference data.

---

> ### Author Response · Authors · 2025-11-22
> **Response to Weaknesses 2 (1/2)**
>
> ## Comment 2.1: The experimental setup uses a batch size of 8, whereas current standard practice in the field uses batch sizes of 32 to 128.
> **Response to Comment 2.1**: We thank the reviewer for pointing this out. However, both theoretical and empirical studies across deep learning, language modeling, and RL/RLHF show that, once the learning rate and related optimizer hyperparameters are properly scaled, the exact batch size has only a secondary effect on the final performance, compared to factors such as total training steps and data size.
> In particular, Smith et al. [1], McCandlish et al. [2],Lee et al. [3], and more recently Marek et al.[4] demonstrate that networks can be trained with batch sizes ranging from 1 up to tens of thousands without loss in test accuracy, provided the learning rate schedule is adapted appropriately.
> Consistent with these findings, many recent and influential LLM alignment and RLHF papers use batch sizes outside the range of 32–128. As shown in follwing table，for example, InstructGPT(Ouyang et al.,2022) uses batch size of 8 in 175B model training .
> Our setting of batch size 8 therefore lies well within common practice.  We also maintain the same experiment setting across method, so that the comparison is fair and primarily driven by algorithmic differences rather than experiment setting.
>
> | Title       | Model                                   | Batch size |
> |------------|------------------------------------------|------------|
> | InstructGPT[5]| 175B                                    | 8          |
> | ReMax[6] memory experiment     | Llama2-7B                               | 8          |
> | MRPO[7]       | Llama2-7B / Mistral-7B / Qwen1.5-7B     | 8          |
> | Logic-RL[8]   | Qwen2.5-7B                              | 8          |
> | GAPO[9]       | Qwen2.5-7B / Qwen2.5-32B                | 8          |
>
> [1]Smith, S. L., Kindermans, P. J., Ying, C., & Le, Q. V. (2017). Don't decay the learning rate, increase the batch size. arXiv preprint arXiv:1711.00489.
>
> [2]McCandlish, S., Kaplan, J., Amodei, D., & Team, O. D. (2018). An empirical model of large-batch training. arXiv preprint arXiv:1812.06162.
>
> [3]Lee, S., He, C., & Avestimehr, S. (2023). Achieving small-batch accuracy with large-batch scalability via Hessian-aware learning rate adjustment. Neural Networks, 158, 1-14.
>
> [4]Marek, M., Lotfi, S., Somasundaram, A., Wilson, A. G., & Goldblum, M. (2025). Small batch size training for language models: When vanilla sgd works, and why gradient accumulation is wasteful. arXiv preprint arXiv:2507.07101.
>
> [5]Ouyang, L., Wu, J., Jiang, X., Almeida, D., Wainwright, C., Mishkin, P., ... & Lowe, R. (2022). Training language models to follow instructions with human feedback. Advances in neural information processing systems, 35, 27730-27744.
>
> [6]Li, Z., Xu, T., Zhang, Y., Lin, Z., Yu, Y., Sun, R., & Luo, Z. Q. (2023). Remax: A simple, effective, and efficient reinforcement learning method for aligning large language models. arXiv preprint arXiv:2310.10505.
>
> [7]Le, H., Tran, Q. H., Nguyen, D., Do, K., Mittal, S., Ogueji, K., & Venkatesh, S. (2025, April). Multi-reference preference optimization for large language models. In Proceedings of the AAAI Conference on Artificial Intelligence (Vol. 39, No. 23, pp. 24375-24383).
>
> [8]Xie, T., Gao, Z., Ren, Q., Luo, H., Hong, Y., Dai, B., ... & Luo, C. (2025). Logic-rl: Unleashing llm reasoning with rule-based reinforcement learning. arXiv preprint arXiv:2502.14768.
>
> [9]Anschel, O., Shoshan, A., Botach, A., Hakimi, S. H., Gendler, A., Baruch, E. B., ... & Medioni, G. (2025, November). Group-Aware Reinforcement Learning for Output Diversity in Large Language Models. In Proceedings of the 2025 Conference on Empirical Methods in Natural Language Processing (pp. 32382-32403).

---

> ### Author Response · Authors · 2025-11-22
> **Response to Weaknesses 2 (2/2)**
>
> ## Comment 2.2: The KL coefficient is crucial for algorithm performance in both DVPO and baseline methods (including PPO and GRPO), and providing the search range would enhance experimental credibility.
>
> **Response to Comment2.2**: We thank the reviewer for this valuable comment. For the KL coefficient hyperparameter,  we set it to 0.05, which is the default choice in the widely used reinforcement learning library TRL, and as shown in foll recently many RLHF works have used this same KL coefficient ([1][2][3][4]).
>
> | Method           | KL coefficient |
> |------------------|----------------|
> | TRL PPO trainer  | 0.05           |
> | [1]              | 0.05           |
> | [2]              | 0.05           |
> | [3]              | 0.05           |
> | [4]              | 0.05           |
>
> To address your concern, we have additionally **conducted an ablation study on the KL coefficient, performing a grid search in the range [0.01,0.05，0.1]**, the results show that our kl coefficient settings are reasonable and all superior to the baseline, we will add it to the revised manuscript.
>
> | KL   | Mt-bench | arena-hard | alpaca-eval |
> |------|----------|------------|-------------|
> | 0.01 | **8.5**      | 37.5       | 41.66       |
> | 0.05 | 7.72     | **39.2**       | **42.59**       |
> | 0.1  | 7.73     | 38.6       | 40.96       |
>
>
> [1]Zhu, B., Sharma, H., Frujeri, F. V., Dong, S., Zhu, C., Jordan, M. I., & Jiao, J. (2023). Fine-tuning language models with advantage-induced policy alignment. arXiv preprint arXiv:2306.02231.
>
> [2]Lin, Y., Lin, H., Xiong, W., Diao, S., Liu, J., Zhang, J., ... & Zhang, T. (2024, November). Mitigating the alignment tax of rlhf. In Proceedings of the 2024 Conference on Empirical Methods in Natural Language Processing (pp. 580-606).
>
> [3]Shen, W., Zhang, X., Yao, Y., Zheng, R., Guo, H., & Liu, Y. (2024). Improving reinforcement learning from human feedback using contrastive rewards. arXiv preprint arXiv:2403.07708.
>
> [4]Sun, H., Chai, Y., Wang, S., Sun, Y., Wu, H., & Wang, H. (2025). Curiosity-Driven Reinforcement Learning from Human Feedback. arXiv preprint arXiv:2501.11463.

---

> ### Author Response · Authors · 2025-11-22
> **Response to Weaknesses 3 & Question**
>
> ## Comment 3.1:The significance of explicitly fitting a value model needs stronger empirical backing. Given that GRPO and RLOO use Monte Carlo-based methods to estimate values, the authors should provide deeper insights into how separately training a value model enables DVPO to outperform these baselines.
>
> **Response to Comment 3.1**: Thank you for the question. The key benefit of explicitly training a value model in DVPO is its ability to provide **fine-grained, token-level value estimates**, which Monte-Carlo methods like GRPO and RLOO cannot provide.
> GRPO/RLOO use a single sentence-level return, giving every token the same credit signal. In contrast, the GVM—trained with TD-learning over prefixes—assigns **different values to different parts of a response**. Our case study (Fig. 3) shows this clearly: decisive reasoning spans receive high values while misleading continuations receive low values, a level of discrimination impossible under sentence-level Monte-Carlo supervision.
>
> This advantage is also reflected in **RewardBench (Table 3)**, where the GVM outperforms scalar reward models on the most challenging subsets (Chat-Hard, Reasoning), precisely where fine-grained value estimation matters most. Finally, across all policy evaluations (Tables 1–2), DVPO consistently outperforms GRPO, demonstrating that finer token-level credit assignment leads to more effective optimization than sequence-level Monte-Carlo returns.
> We will add this clarification to the final version of the paper.
>
> ## Comment 3.2: Furthermore, the comparison in Table 6 is unfair to GRPO regarding response generation time
>
> **Response to Comment 3.2**: We thank the reviewer for pointing this out. We agree that wall-clock "generation time" can be confounded by implementation details and parallelization strategies, and thus is not the most appropriate metric here. In the revised version, we will replace the "response generation time" entry in Table 6 with a clearer measure of **computational cost**. Conceptually, compared to PPO/DVPO, GRPO requires sampling $n$ responses per prompt within each group, which leads to roughly $n$-times more forward passes through the policy (i.e., on the order of $n \times$ batch size in total responses). This multi-sample generation is also the dominant source of GRPO's training cost. While the *actual* wall-clock time may vary depending on GPU parallelism and implementation, the intrinsic algorithmic compute required by GRPO scales linearly with $n$, whereas PPO/DVPO only need one response per prompt. We will clarify this point in the text and adjust Table 6 accordingly to focus on computational cost rather than raw generation time.
>
> ## Question 1: Have you considered incorporating Bradley-Terry reward modeling in the pre-training of value models? This is the most standard approach to reward modeling, as human feedback typically comes in pairwise comparisons. Simply assigning -1 and +1 rewards does not reflect the actual reward modeling procedure.
>
> **Response to Questions**: Yes, we have considered Bradley–Terry (BT) style reward modeling when pre-training the value model. In fact, as discussed in Appendix G *Analysis of Rewards Assigned* and summarized in **Table 9**, we explicitly compare two variants: (i) pre-training the value model on scalar rewards produced by a BT reward model, and (ii) directly pre-training it on pairwise preferences encoded as $-1$ / $+1$ labels. As shown in Table 9, the overall performance of these two variants is very similar, and the preference-based training with $-1$ / $+1$ labels is slightly better.
>
> One plausible explanation is that the BT reward model introduces an additional source of approximation error: since the reward model itself does not achieve 100% accuracy even on the training pairs, its scalar outputs inevitably contain some noise, which is then further propagated into the value model. By contrast, directly training on the original pairwise comparisons avoids this extra modeling  and thus may lead to a cleaner supervision signal.

---

> > ### Comment · Reviewer_PVQp · 2025-11-26
> >
> > Thank you for your detailed response and diligence during the rebuttal stage. After thoroughly reviewing your responses, I find that my major concerns have not been adequately addressed. Below are my detailed comments:
> > ### 1. The philosophy behind DVPO's advantage is unclear
> > while DVPO pre-trains the value model and keep it fixed, this indeed avoids the instability issue of actor-critic. However, the true advantage estimation requires access to $V^{\pi_\theta}$ for on-policy value estimation. Using multi-policy trajectories cannot help DVPO overcome such a weakness.
> >
> > ### 2. Positioning DVPO relative to stablized PPO
> >
> > I acknowledge this statement: *Token-level advantage for more accurate credit assignment*.
> >
> > However, this raises a more fundamental question about the paper's contribution: PPO also performs token-level advantage optimization. If DVPO's primary advantage is resolving PPO's instability, how does it compare against stabilized PPO variants?
> >
> > Several established techniques stabilize PPO training for RLHF, including: 1)freezing the actor/policy while performing online warm-up of the value model, 2) pre-training the value model on offline trajectories. Both approaches use pre-trained reward models and combine warm-up (offline or online) with subsequent online alignment. How is your method more preferable over these approaches?
> >
> > ### 3. Conversion of pairwise preference labels to absolute scalar rewards (-1, +1)
> >
> > I remain concerned about the conversion of preference signals to -1/+1 rewards. Preference labels encode relative pairwise comparisons, whereas converting them to absolute values of +1 and -1 fundamentally changes the learning objective from preference modeling to something closer to imitation learning (analogous to GAIL [Ho & Ermon, 2016]).
> >
> > This raises questions about whether DVPO is truly performing preference alignment or rather imitating good responses from the policy pool in an online setting.
> >
> > ### 4. Regarding batch size
> > I appreciate your response, but the justification for using a batch size of 8 remains insufficient. Influential RLHF papers consistently use batch sizes of 64-128:
> > * Rafailov et al. (2023) - Direct Preference Optimization
> > * Xu et al. (2024) - "Is DPO Superior to PPO for LLM Alignment?"
> > * Zhong et al. (2024) - "DPO Meets PPO"
> > * Shao et al. (2024) - DeepSeekMath
> > * Yu et al. (2025) - DAPO
> >
> > ### 5. Hyperparameter Tuning on KL coefficients $\beta$
> > Your ablation study on KL coefficients shows that DVPO achieves optimal results at different β values for different benchmarks (β=0.01 for MT-Bench, β=0.05 for Arena-Hard and AlpacaEval). To fairly demonstrate DVPO's advantages, I strongly encourage you to present reward/KL Pareto frontiers for both DVPO and baseline methods across multiple hyperparameter settings. This is the standard approach for evaluating alignment methods and would provide clearer evidence of whether DVPO offers genuine improvements.
> >
> > ---
> >
> > Given these outstanding concerns, my score remains unchanged.

---

> ### Author Response · Authors · 2025-11-28
> **Reply to Rebuttal Response (1/3)**
>
> ## comment1 ：The philosophy behind DVPO's advantage is unclear
>
> **Response to Comment1:** Thank you for the question. We would clarify property of the standard human-labeled reward RLHF setting, where once a reward model is trained from human data, it is fixed and all subsequent optimization, without any further access to ground-truth human feedback. Existing RLHF methods, such as PPO, GRPO, RLOO, and ReMax, receives rewards only from this fixed reward model (except for verified-reward environments like programmatically checkable math or code). In this regime, no method can obtain true on-policy estimates with respect to the underlying human reward function; all operate on trajectories scored by the same fixed reward model. DVPO’s advantage is therefore not based on a different source of reward, but on how it uses the available offline data and stabilizes training. We exploit multi-policy trajectories to train a global value model that provides fine-grained, token-level return-to-go estimates, and then freeze this value surface during RL to avoid the instabilities of online actor–critic coupling. Our experiments show that the pretrained value model provides finer token-level credit (Fig. 3, Table 3), and when used in DVPO, leads to consistently stronger policy performance than GRPO/PPO/Remax (Tables 1–2). This reinforces our main contribution: **pretraining the value model is a more effective way to leverage  than pretraining only a reward model.**
>
> ## Comment2: Positioning DVPO relative to stablized PPO
>
> **Response to Comment2** :
> Thank you for the question. Our work introduces a **value-pretraining perspective** for human-labeled reward RLHF.In **Section 3.4**, we provide a formal analysis showing that information-level equivalence between reward-model-based value learning and value pretraining. However, PPO-style methods require alternating optimization of the policy and value model during training, this alternating optimization introduces additional complexity and instability into the training process. Our experiments also reflect this: the pretrained value model trained with TD-learning avoids the two major weaknesses of existing RLHF baselines—critic drift in PPO-style actor–critic RL and the coarse sequence-level Monte-Carlo credit assignment used in GRPO/RLOO.
>
>
> For stabilized PPO variants, to the best of our knowledge, there is not yet **a published, peer-reviewed, or reproducible** RLHF algorithm that implements the specific PPO variants you describe and is evaluated on common RLHF benchmarks. In contrast, DVPO adopts a clear, reproducible, and well-motivated design point: (1) a value model pretrained entirely offline on data, (2) frozen during policy optimization to avoid critic drift, and (3) used as a fixed critic providing fine-grained signals unavailable from reward models or Monte-Carlo estimators.
>
> This decoupled formulation is both theoretically principled and empirically validated by our results. Together, these findings support our main claim: pretraining a value model, rather than a reward model for RLHF.

---

> ### Author Response · Authors · 2025-11-28
> **Reply to Rebuttal Response (2/3)**
>
> ## Comment3:Conversion of pairwise preference labels to absolute scalar rewards (-1, +1)
>
> **Response to Comment4**:
> Thank you for the concern. We believe this concern may stem from a misunderstanding and clarify below.
>
> First, as stated in Section 4.1 (line 281-285) of our paper, our *main* experiments are conducted using scalar reward scores.  The UltraFeedback dataset we use already provides a scalar reward for each response.  Therefore, even if there were issues with converting preference labels to $(-1, +1)$, this would not affect the conclusions of our main experiments. We will clarify this point more explicitly in Section 4.1.
>
> Second, we only convert preference data to $(-1, +1)$ in an **extend** experiment to evaluate DVPO on *pure preference* datasets that do not contain scalar rewards.  Concretely, we convert UltraFeedback into a preference dataset following the  `HuggingFaceH4/ultrafeedback_binarized`, and then assign reward $+1$ to the chosen response and $-1$ to the rejected response.  To put it another way, mapping pairwise preferences to scalar rewards is a widely used and well-studied practice in RL/RLHF, and it does *not* turn the problem into imitation learning.  On the theoretical side, the ICLR 2025 oral paper[1] argues that what really matters for preference-based reward models is **order consistency**: as long as the learned reward preserves the same ordering as the true utility (i.e., $r(A) > r(B)$ whenever humans prefer $A$ over $B$), then any strictly monotone transform of the reward leads to essentially the same optimization behavior for response selection.  This perspective is compatible with using coarse scalar encodings such as assigning larger scores to preferred than to rejected responses (e.g., $+1$ vs. $-1$ at the pair level), rather than modeling exact comparison probabilities. On the practical side,  such as KTO (ICML 2024)[2] and BCO (ACL 2025)[3] shows that learning from **binary feedback alone** can match or even surpass standard preference-based methods like RLHF and DPO. Together, these results indicate that using binary or scalar feedback remains a preference-alignment procedure rather than reducing to imitation learning.
>
>
> [1]Sun, H., Shen, Y., & Ton, J. F. (2024). Rethinking bradley-terry models in preference-based reward modeling: Foundations, theory, and alternatives. arXiv preprint arXiv:2411.04991.
>
> [2]Ethayarajh, K., Xu, W., Muennighoff, N., Jurafsky, D., & Kiela, D. (2024). Kto: Model alignment as prospect theoretic optimization. arXiv preprint arXiv:2402.01306.
>
>
> [3]Jung, S., Han, G., Nam, D. W., & On, K. W. (2025, July). Binary classifier optimization for large language model alignment. In Proceedings of the 63rd Annual Meeting of the Association for Computational Linguistics (Volume 1: Long Papers) (pp. 1858-1872).
>
> ## Commment 4: Regarding batch size
>
> **Response to Comment 4:** Thank you for the comment. To clarify, we are not arguing that batch sizes of 64–128 are inappropriate, and these settings are perfectly valid in certain RLHF pipelines. Our point is simply that batch size 8 is equally standard and widely used in RLHF, especially in PPO-style or rollout-based training. Influential RLHF systems such as **InstructGPT, ReMax, MRPO, Logic-RL, and GAPO** all use batch size = 8 for LLM reinforcement learning. These works demonstrate that batch size 8 is a normal and accepted configuration in the community.
>
> We would also like to gently clarify that the papers cited by the reviewer do not support the claim that “influential RLHF work consistently uses batch sizes of 64–128.” For example, DeepSeekMath uses a batch size of 1024, DAPO uses 512, and DPO Meets PPO uses 128–512. These values illustrate that practical batch-size choices vary widely across RLHF pipelines and are largely dictated by GPU memory, rollout cost, and engineering constraints, not by a fixed “correct” range. In addition, given that the papers cited by the reviewer themselves use batch sizes of 128–1024 rather than 64–128, it is unclear why 64–128 is presented as the “standard” or “correct” range by the reviwer. These citations do not support that claim.
>
> Therefore, it would not be accurate to regard 64–128 as the only appropriate setting while considering batch size 8 insufficient. What matters for scientific comparison is fairness, and in our experiments, all methods (DVPO, PPO, GRPO, Remax,etc) use exactly the same batch size, ensuring that differences arise from algorithmic factors rather than hyperparameters.
> We kindly ask the reviewer to evaluate our setting in this broader and more representative context: **batch size 8 is a common RLHF configuration**, and using it consistently across all baselines yields a fair and meaningful comparison.

---

> ### Author Response · Authors · 2025-11-28
> **Reply to Rebuttal Response (3/3)**
>
> ## Comment 5:  Hyperparameter Tuning on KL coefficients
>
> **Response to Comment 5**: Thank you for the thoughtful suggestion.
>
> For **baseline methods**, our philosophy is that the original papers are responsible for hyperparameter tuning, and downstream work should adopt their recommended “best-practice” settings. Concretely, we collected the recommended KL coefficients used in the corresponding implementations, summarized in the following table:
>
> | Method | \(kl\) | Source              | \(kl\) (Our Paper Table 1/2) |
> |--------|--------|---------------------|--------------------|
> | PPO    | 0.05   | TRL implementation  | 0.05               |
> | ReMax  | 0.10   | [1]                 | 0.05               |
> | DPO    | 0.05   | [1]                 | 0.05               |
> | GRPO   | 0.04   | [2]                 | 0.05               |
>
>
> In our **main results (Table 1 and Table 2)**, to ensure a *fair and controlled* comparison, we set the KL coefficient of *all* methods to \(kl = 0.05\). This value lies in the recommended range for most baselines and avoids giving DVPO extra tuning budget that baselines do not enjoy. For DPO, we use the optimal KL of 0.05 recommended in [1]. For GRPO, the difference between \(kl = 0.04\) and \(kl = 0.05\) is extremely small, and aligning GRPO to \(kl = 0.05\) helps make the comparison across methods more interpretable and consistent.
>
> Motivated by your comment, we additionally ran an experiment where **ReMax** uses its recommended \(kl = 0.10\). The results are:
>
> | \(kl\) | MT-Bench | Arena-Hard | Alpaca-eval |
> |--------|----------|------------|------------|
> | 0.10   | 7.58     | 29.1       |36.23
> | 0.05   | 7.62     | 30.8       | 36.04
>
> This shows that changing ReMax from \(kl = 0.05\) to \(kl = 0.10\) does **not** systematically improve its performance: MT-Bench and Arena-Hard slightly **decrease**. Overall, these differences are modest, and our key qualitative conclusions about DVPO’s advantage are unchanged. We will include this additional ablation and a clarification of our hyperparameter choices in the revised version.
>
> Within our computational budget, we therefore: (i) adopt same KL coefficients in baselines and DVPO, and (ii) adopt published best-practice KL coefficients for baselines, plus the new ReMax \(kl = 0.10\) experiment above to confirm that our conclusions are not an artifact of our choice of \(kl\). We hope this addresses your concern.
>
> [1] Li, Z., Xu, T., Zhang, Y., Lin, Z., Yu, Y., Sun, R., & Luo, Z. Q. (2023). Remax: A simple, effective, and efficient reinforcement learning method for aligning large language models. arXiv preprint arXiv:2310.10505.
>
> [2] Shao, Z., Wang, P., Zhu, Q., Xu, R., Song, J., Bi, X., ... & Guo, D. (2024). Deepseekmath: Pushing the limits of mathematical reasoning in open language models. arXiv preprint arXiv:2402.03300.

---

### Official Review · Reviewer_q6ih · 2025-10-31

**Soundness:** 3
**Presentation:** 3
**Contribution:** 3
**Rating:** 6
**Confidence:** 3

**Summary:**

The paper proposes Decoupled Value Policy Optimization (DVPO) for RLHF: instead of the standard reward-model → online critic → policy pipeline, it pretrains a single Global Value Model (GVM) on the same offline preference data and freezes it as a universal critic to guide policy learning. The problem being solved is the instability and compute overhead of actor–critic RLHF (critic drift, high variance from sequence-level rewards, and the need to run multiple models in training).

Key claims:
(Theoretical) With fixed feedback (no new rewards), reward-model pretraining + value estimation is informationally equivalent to direct value pretraining; policy gradients differ by at most a bounded term κ that vanishes as approximation errors go to 0 (Lemma 3.1). A corollary states DVPO preserves PPO-style local convergence under KL regularization.
(Algorithmic) DVPO eliminates online critic training, using a frozen, token-level value signal (GVM) for advantages, thereby avoiding critic drift and reducing variance while retaining on-policy updates.
(Empirical) On MT-Bench, Arena-Hard, and Alpaca-Eval, DVPO matches or surpasses PPO/GRPO/ReMax/DPO across Llama-3B/8B and Mistral-7B settings (Tables 1–2), and shows improved compute efficiency (≈40–50% lower memory, ≈35% faster training in the base setting; Table 7).

**Strengths:**

Clear, impactful simplification: Recasting RLHF as policy-only optimization with a pretrained value is elegant and practically meaningful.
Solid theory–practice bridge: The equivalence lemma and convergence corollary directly justify the algorithmic design.
Consistent empirical gains & breadth: Improvements across models/benchmarks in both base and instruction settings; strong compute-efficiency results.
Token-level credit & interpretability: Concrete examples demonstrate fine-grained attributions absent in reward-only methods.
Reusability/generalization: A policy-agnostic critic simplifies reuse across models and appears robust vs. actor–critic critics on distribution shift.

**Weaknesses:**

Fixed-feedback assumption (no new rewards). DVPO’s theory and setup hinge on no additional reward during training; if limited online human feedback arrives, the frozen GVM cannot adapt.
Q: If limited online feedback becomes available, could you support periodic GVM refresh (semi-online DVPO) while retaining the stability guarantee, and what parts of Lemma 3.1 / the corollary would need to change?

Scope of evaluations. Benchmarks are mainstream chat-style; tougher code/maths/long-horizon tasks may stress token-value fidelity.
Q: Any results (or planned) on reasoning-heavy suites to test structure-aware value estimation?

Stability evidence presentation. Paper claims reduced critic drift and smoother training but lacks side-by-side variance/curve plots quantifying it.
Q: Can you add curves (PPO vs DVPO) and a critic-policy disagreement metric (e.g., advantage-distribution KL) to concretely show drift reduction?

**Questions:**

seen in the weakness section.

Table 1 (Base setting). Please include #prompts and #rollouts per prompt used during RL for each method (PPO/GRPO/ReMax/DVPO), and whether generation lengths and KL targets were matched across methods (important for fairness).

Compute tables (Table 6 vs Table 7). The narrative states ~40% memory and ~35% time reductions, while Table 7 gives concrete GB/seconds for specific setups. Please reconcile the headline percentages with the numbers in Table 7

---

> ### Author Response · Authors · 2025-11-22
> **Response to Weaknesses 1& Weaknesses 2**
>
> ## Weaknesses 1: Fixed-feedback assumption (no new rewards).
>
> **Response to Comment 1**: We appreciate your insightful question — this is indeed an interesting topic.
>
> In this work, we focus on a more classical setting — the one adopted by most RLHF optimization frameworks — where the key question is how to optimize the model after a fixed amount of preference data has been collected. Existing approaches typically follow a two-stage paradigm: first training a reward model, and then using this reward model to provide supervision signals during the reinforcement learning process (e.g., PPO, GRPO, etc.). Therefore, our theoretical analysis (Lemma 3.1 and its corollary) is built upon the assumption of a fixed GVM learned entirely offline, ensuring a provable stability guarantee under that setting.
>
> Extending the framework to support periodic GVM refresh or a semi-online DVPO variant is an important direction for future work. We envision addressing it by incorporating a lightweight periodic update of the GVM using newly collected preference data, while controlling the update frequency and drift magnitude to ensure quasi-stationarity. Specifically, the GVM could be refreshed every \(K\) policy updates using a small batch of recent preference comparisons, followed by freezing during subsequent updates. This resembles the semi-online preference optimization paradigm, which has been successfully adopted in recent RLHF literature to balance adaptability and stability[1].
>
> To support such an extension, we would carefully revisit Lemma 3.1 by relaxing the fixed-feedback assumption. In particular, we could introduce a bounded-drift condition on the periodically refreshed GVM: for example, requiring that the expected deviation between consecutive GVMs be bounded by a constant $\delta$. Under this relaxed condition, the stability result in Lemma 3.1 can be preserved up to an additional $\delta$-dependent error term. We also outline how the preference error bound can be generalized to $\kappa(\epsilon_R, \epsilon_Q + \delta)$, where $\delta$ quantifies the impact of GVM drift over time. This extension would provide a principled path toward supporting semi-online DVPO with theoretical guarantees.
>
> For completeness, we provide the following reformulated version of Lemma 3.1:
>
> ---
>
> **Lemma 3.1 (Extended: Stability under Periodically Refreshed GVM).**
> Assume the GVM $\hat{Q}_{\phi_t}$ is refreshed every $K$ policy updates using new preference data, and is kept fixed within each refresh interval. Suppose the deviation of the refreshed GVM from the previous version is uniformly bounded, i.e.,
>
> $
> \forall t,\quad \sup_{\pi \in \Pi} \left| \mathbb{E}_{x\sim d^\pi} [\Delta \hat{Q}_t(x,\pi)] \right| \leq \delta.
> $
>
> where $\Delta \hat{Q}t(x,\pi)$ denotes the difference between the current and previous GVM estimates, i.e., $\hat{Q}{\phi_t}(x,\pi) - \hat{Q}{\phi_{t-K}}(x,\pi)$.Then, under Assumptions (1)–(2) in the original Lemma, the preference error satisfies:
>
> $
> \text{PREF}(\pi) \leq \kappa\left(\epsilon_R, \epsilon_Q + \delta \right),
> $
>
> where $\epsilon_Q$ denotes the maximum deviation of $\hat{Q}_{\phi_t}$ from $Q^*$ at each refresh point. The constant \(\kappa\) remains the same as in the original statement.
>
> ---
>
> We thank you again for pointing out this promising line of research.
>
> [1]Lanchantin, J., Chen, A., Lan, J., Li, X., Saha, S., Wang, T., ... & Kulikov, I. (2025). Bridging Offline and Online Reinforcement Learning for LLMs. arXiv preprint arXiv:2506.21495.
>
> ## Weakness 2: Scope of evaluations.
>
> **Response to Weakness 2**: We agree that structure-aware value estimation should ideally be evaluated on reasoning-heavy tasks such as math, code, and long-horizon dialogue.  In our current setup, both the training data and evaluation suites already contain a non-trivial proportion of such problems.  Specifically, our mixed training corpus includes math/code and  reasoning samples, and the benchmarks used in the paper (Mt-bench/Arena-Hard/AlpacaEval 2) also contain math and coding items.  In addition, RewardBench includes a dedicated “reasoning” subset, for which we have selected the corresponding data to compare GVM against baselines.
>
> Table 3: GVM vs ScalarRM on reward-bench (Reasoning)
>
> | Model      | Method   | Reason. |
> |------------|----------|---------|
> | Llama3 8B  | GVM      | 79.2    |
> |            | ScalarRM | 65.4    |
> | Mistral 7B | GVM      | 64.4    |
> |            | ScalarRM | 60.3    |
>
>
> To further address the reviewer’s concern, we are running additional experiments that focus explicitly on math/code and other reasoning-heavy subsets, and will report these results in the revised version.

---

> > ### Author Response · Authors · 2025-11-26
> > **Additional results about reasoning-heavy tasks**
> >
> > We conducted an additional evaluation analysis of DVPO on the reasoning-heavey subsets of MT-Bench. As shown in following table, which separates questions into math, code, and general reasoning categories, DVPO achieves the highest scores on both the reasoning and coding subsets, and remains highly competitive on the math subset. These findings indicate that our method  yield clear improvements on reasoning-heavy tasks. Thank you for your insightful suggestions, we will update this result to the pdf. We hope we can address your concerns.
> >
> > | Method       | Reasoning | Math | Code | Overall |
> > |-------------|-----------|------|------|---------|
> > | instruction | 5.45      | 5.75 | 5.25 | 6.90    |
> > | ReMax       | 6.00      | 6.20 | 6.05 | 7.62    |
> > | DPO         | 5.80      | 6.10 | 5.65 | 7.01    |
> > | GRPO        | 5.25      | **7.25** | 5.45 | 7.53    |
> > | PPO         | 6.20      | 7.05 | 6.05 | 7.55    |
> > | DVPO        | **6.30**      | 7.15 | **6.10** | **7.72**    |

---

> > > ### Author Response · Authors · 2025-11-28
> > > **Thank you for your valuable feedback**
> > >
> > > Thank you for your valuable feedback and the opportunity to address your concerns. We hope our responses have provided clarity and further strengthened the work. We would sincerely appreciate your consideration of a higher score.

---

> ### Author Response · Authors · 2025-11-22
> **Response to Weakness 3 & Questions**
>
> ## Weakness 3: Stability evidence presentation
> **Response to Weakness 3**: We thank the reviewer for this helpful suggestion.  Unlike PPO, in DVPO the Global Value Model (GVM) is **pre-trained and kept fixed** during policy optimization.  As a result, the learning target for the policy does not drift over the course of training, in contrast to the continuously-updated critic in standard actor–critic methods.
>
> This effect can already be observed in our current empirical results.  As reported in Table 4 and Table 5, the GVM provides higher-quality supervision and more accurate value estimates compared to the on-policy critic, which directly supports our claim that DVPO benefits from a more reliable value signal.  Furthermore, in Fig. 4 we present side-by-side training curves for PPO and DVPO under matched hyperparameters on LLaMA-3 3B and LLaMA-3 8B.  These curves show that DVPO achieves **smoother and more stable** learning dynamics, with less fluctuation across updates than PPO, consistent with reduced critic drift.
>
> ##Question 1: Table 1 (Base setting)
>
> **Response to Question1**: We thank the reviewer for this helpful suggestion.  We use the same dataset of training prompts and match all key hyperparameters across PPO, GRPO, ReMax, and DVPO to ensure a fair comparison.   Concretely, we use a batch size of 8, a maximum generation length of 1024 tokens, and a KL coefficient of 0.05 for all methods.   For the Monte Carlo–based GRPO, we sample 5 rollouts per prompt.   In the revised version, we will more explicitly report the training details.
>
> ##Question 2: Compute tables (Table 6 vs Table 7).
>
> **Response to Question 2**: We thank the reviewer for catching this inconsistency and for the opportunity to clarify.   Table 7 reports the **actual peak GPU memory usage and per-step wall-clock time** measured during training under our practical RLHF setup.   Note that this setup is slightly different from the idealized time configuration discussed in the introduction: for the 3B model we use DeepSpeed Zero-1, while for the 8B model we use Zero-3 due to OOM constraints.   These different parallelization strategies can make the *observed* GB/second numbers deviate from the simple theoretical compute ratios.
>
> If we convert the numbers in Table 7 into relative percentages, DVPO reduces peak memory consumption by roughly 23–34% and per-step time by about 32–44% compared to PPO, but we agree that the current wording is not sufficiently precise when read side by side with Table 7.
>
> As shown in Table 6, the source of these savings is clear: compared to PPO, DVPO removes the need to **train an additional value model with full activation checkpoints**.   In the standard RLHF setting, the critic (value model) and the policy typically have comparable parameter sizes and both require activation storage and backpropagation, which leads to substantial memory and compute overhead.   In DVPO, the Global Value Model (GVM) is pre-trained and frozen, so only the policy model is actively updated, reducing both activation memory and optimization cost.
>
> In the revised version, we will (i) add the corresponding percentage reductions to the tables, and (ii) adjust the main-text wording to match the exact values (e.g., “around 30–40% memory and 30–45% time reductions, depending on the backbone and parallelization strategy”) so that the narrative is fully consistent with the reported numbers.
>
> Revised Table 7: Computational results under the **Base setting**.
> For the 3B model, we use DeepSpeed Zero-1; for the 8B model, we use Zero-3 due to OOM issues.
> Numbers in parentheses for DVPO denote percentage reduction relative to PPO.
>
> | Method | Llama3-8B-Base Mem (GB) | Llama3-8B-Base Time (s) | Llama3.2-3B-Base Mem (GB) | Llama3.2-3B-Base Time (s) |
> |--------|-------------------------|-------------------------|---------------------------|---------------------------|
> | PPO    | 78.96                   | 70.24                   | 41.43                     | 23.00                     |
> | ReMax  | 61.23                   | 65.96                   | 26.87                     | 44.13                     |
> | GRPO   | 64.35                   | 57.93                   | 28.20                     | 22.17                     |
> | DVPO   | 60.49 (-23.4%)          | 48.00 (-31.7%)          | 27.50 (-33.6%)            | 13.00 (-43.5%)            |

---

### Author Response · Authors · 2025-11-28
**Global Response**

We are deeply grateful to all reviewers [R1 (q6ih), R2 (PVQp), R3 (sBiL)] and the AC for their careful evaluation of our manuscript and for their thoughtful and constructive feedback.

In this manuscript, we propose a new perspective on RLHF via **value pretraining**. We theoretically analyze the information-level equivalence between pretrain reward and pretrain value, and empirically validate the effectiveness of value pretraining through extensive experiments.  We are delighted that the reviewers recognized **the importance of the problem addressed**[R1, R2, R3], appreciated the **well-supported motivation** [R1, R3], found our **theoretical analysis clear** [R1, R2, R3], and considered our experiments comprehensive with **encouraging results** [R1, R2, R3].

Based on the reviewers’ feedback, we have carefully revised the manuscript and uploaded a new version for review. All changes are highlighted in blue. Below we summarize the key updates:

- **Section 5.2 (lines 404–406) and Section 6.2 (lines 447–449):**  We further analyze and clarify the global property of the GVM and the underlying reasons for the observed performance gains. [R2]

- **Limitations (lines 486–489):**  We discuss the very promising future work of applying DVPO to online or semi-online RLHF scenarios. [R1, R3]

- **Appendix C (lines 803–806):**  We provide more detailed training hyperparameters for DVPO and baselines. [R1, R2]

- **Appendix F and Tables 6 & 7:**  We use more precise and technically accurate descriptions of DVPO’s computational efficiency. [R1, R2]

- **Appendix H:**  We conduct ablation study by reporting DVPO’s performance under different KL coefficients, which demonstrates that our chosen setting is reasonable and comparable across methods. [R2]

- **Additional results:**
  1. To analyze DVPO’s effectiveness on reasoning-heavy tasks, we further analysis DVPO on different subsets of MT-Bench. The results show that DVPO still outperforms (or is comparable to) the baselines in reasoning-heavy scenarios. [R1]
  2. To ensure the comparability and transparency of our experiments, we conduct experiments under two settings: (i) all baselines are trained with the same hyperparameters; and (ii) each baseline is trained with the best-practice in its original paper. DVPO consistently performs well under both settings. [R2]

---

### Meta-Review · Area_Chair_YBtG · 2026-01-06

**Summary:**

DVPO proposes simplifying RLHF by pretraining a single Global Value Model (GVM) on offline preference/reward data and freezing it as a universal critic, avoiding online critic training and the associated drift/variance/compute overhead. Reviewers’ main decision-driving concerns were whether the fixed-feedback assumption limits applicability (no mid-training new feedback), whether evaluations and reporting are sufficiently fair/complete (matched rollouts, lengths, KL, clearer compute numbers), and whether stability claims are supported by direct evidence (curves/metrics), with one reviewer additionally questioning the conceptual positioning (global value vs on-policy advantage) and standard RLHF evaluation practices.
The approach is a practically meaningful simplification with solid theory in the fixed-feedback regime, strong benchmark results across multiple models, and credible efficiency gains; the rebuttal strengthened scope/fairness/compute/stability evidence. The remaining dissent centers on positioning versus stabilized PPO variants and evaluation norms (Pareto sweeps), which should be addressed in the final version but do not outweigh the contribution and overall positive reception.

**Reviewer Concerns:**

The rebuttal largely addressed requests for broader reasoning-focused evidence (RewardBench reasoning + MT-Bench reasoning/math/code breakdown), clarified fairness details (matched key hyperparameters, GRPO rollout count), reconciled compute headline claims with Table 7 via revised percentages and parallelization explanations, and pointed to PPO-vs-DVPO training curves plus better value-quality tables to support stability. Still outstanding are one reviewer’s core objections: unclear advantage philosophy vs stabilized PPO variants, concerns about preference-to-scalar conversion (despite authors clarifying main experiments use scalar rewards), and the lack of KL–reward Pareto/frontier-style sweeps that are often expected for alignment method comparisons.

**Reviewer Scores:**

Estimated updates if fully engaged: q6ih likely 6→7 (most requested clarifications/added results were provided); sBiL likely stays 8→8 (already strongly positive); PVQp likely stays 2→2 (reviewer explicitly states concerns remain unresolved after rebuttal); LCEj withdrew (no score).

---

### Decision · Program_Chairs · 2026-01-26

Accept (Poster)